# Biogenic non-crystalline U(IV) revealed as major component in uranium ore deposits

Amrita Bhattacharyya[1,†], Kate M. Campbell[2], Shelly D. Kelly[3], Yvonne Roebbert[4], Stefan Weyer[4], Rizlan Bernier-Latmani[5] & Thomas Borch[1,6]

Historically, it is believed that crystalline uraninite, produced via the abiotic reduction of hexavalent uranium (U(VI)) is the dominant reduced U species formed in low-temperature uranium roll-front ore deposits. Here we show that non-crystalline U(IV) generated through biologically mediated U(VI) reduction is the predominant U(IV) species in an undisturbed U roll-front ore deposit in Wyoming, USA. Characterization of U species revealed that the majority (~58-89%) of U is bound as U(IV) to C-containing organic functional groups or inorganic carbonate, while uraninite and U(VI) represent only minor components. The uranium deposit exhibited mostly $^{238}$U-enriched isotope signatures, consistent with largely biotic reduction of U(VI) to U(IV). This finding implies that biogenic processes are more important to uranium ore genesis than previously understood. The predominance of a relatively labile form of U(IV) also provides an opportunity for a more economical and environmentally benign mining process, as well as the design of more effective post-mining restoration strategies and human health-risk assessment.

[1] Department of Soil and Crop Sciences, Colorado State University, Fort Collins, Colorado 80523-1170, USA. [2] US Geological Survey, Boulder, Colorado 80303, USA. [3] EXAFS Analysis, Bolingbrook, Illinois 60440, USA. [4] Institut für Mineralogie, Leibniz Universitat Hannover, Hannover D-30167, Germany. [5] Environmental Microbiology Laboratory, École Polytechnique Fédérale de Lausanne, Lausanne CH-1015, Switzerland. [6] Department of Chemistry, Colorado State University, Fort Collins, Colorado 80523-1872, USA. † Present address: Lawrence Berkeley National Laboratory, Berkeley, California 94720, USA. Correspondence and requests for materials should be addressed to T.B. (email: Thomas.Borch@colostate.edu).

Typical sandstone-hosted uranium (U) roll-front deposits are formed in confined aquifers at a redox front of oxidized groundwater reacting with unaltered reduced aquifer materials, often associated with organic detritus[1,2]. Roll-front deposits are an economically important ore for *in situ* recovery (ISR) mining, whereby U is mined via solubilization of the ore by a leach solution injected in the subsurface. The reduction mechanism for $U^{(VI)}$ in roll-front deposits has long been thought to be abiotic electron transfer by redox active minerals (pyrite and mackinawite), aqueous sulfide and/or other reactive sulfur species[1–5]. Other possible abiotic $U^{(VI)}$ reduction mechanisms observed in low-temperature environments include reaction with adsorbed $Fe^{(II)}$, structural $Fe^{(II)}$ in clays and reduced organic functional groups (thiols)[1,6–8]. Previous mineralogy studies of sandstone-hosted roll-front deposits have identified crystalline uraninite and coffinite as the dominant U-bearing minerals[1,2]. The role of bacteria has been poorly understood and was thought to be limited to bacterial production of reducing agents (for example, sulfide) that play a key role in U deposit formation[1]. Over the past decade, with the advent of synchrotron radiation-based spectroscopy, the presence of short-range ordered nano-particulate uraninite, presumed to result from enzymatic reduction of $U^{(VI)}$, has frequently been reported in a variety of U-reducing environments[9–15]. More recently, a non-uraninite $U^{(IV)}$ species, commonly termed non-crystalline $U^{(IV)}$ (but also referred to as 'monomeric $U^{(IV)}$', 'mononuclear $U^{(IV)}$' and 'molecular $U^{(IV)}$') was reported to form in the presence of biofilms and specific inorganic ligands in laboratory-based studies and field settings[16–23]. A study by Bargar *et al.*[16] showed the formation of non-crystalline $U^{(IV)}$ in a shallow U-contaminated aquifer during microbial bioremediation with acetate as an electron donor. The authors hypothesized that the reduction occurred through a combined biotic–abiotic pathway mediated by biogenic mackinawite (FeS) followed by complexation of the resulting $U^{(IV)}$ by extracellular polymeric substances. Further, non-crystalline $U^{(IV)}$ species with a structure lacking the features of a U–U pair correlation were observed within a sediment sample collected from an ISR U mine following 5 years of post-mining restoration[24]. Finally, the heavy U isotope signature identified in groundwater and uranium ore concentrates from low-temperature sandstone ore formations suggests that biological reduction maybe a dominant process in those systems[25–27]. Thus, these new developments in environmental speciation of reduced U and U isotope interpretations call for re-evaluation of the prevailing model for U ore formation in roll-front deposits in order to optimize ore exploration and exploitation, as well as mine restoration. The aforementioned studies targeted either microbial $U^{(VI)}$ reduction within a laboratory setting or bioremediation of $U^{(VI)}$ within contaminated shallow aquifer systems, which are very different from ore formation in roll-front deposits. Our study aims to characterize U naturally present in an undisturbed roll-front deposit and therein lies the novelty of this research.

Here, we investigate formation mechanism and $U^{(IV)}$ species naturally present in undisturbed roll-front U deposits in Wyoming, USA (Supplementary Figs 1 and 2). We observe that non-crystalline $U^{(IV)}$ is a dominant U species and hypothesize that it is primarily formed via direct enzymatic $U^{(VI)}$ reduction. These deposits occur in three superposed arkosic sandstone units separated by thin beds of mudstone, interbedded shales and high-carbonate zones[3], and are ~3 million years old according to U-Pb dating[28]. Based on X-ray diffraction (XRD), a previous study at a location situated 10 km from our study site identified pitchblende, uraninite and coffinite as the main crystalline U-bearing minerals[29]. However, non-crystalline $U^{(IV)}$ lacks long-range order, thus eluding identification via standard

mineralogical characterization[17]. We combine advanced techniques in order to fully probe the potential presence of biogenic non-crystalline $U^{(IV)}$ in an unmined ore zone located 200 m below ground surface (m-bgs): X-ray absorption spectroscopy (XAS) was used to determine the valence state and the average local atomic coordination environment of U; sequential extractions were used to determine the sediment fractions hosting U (refs 30,31); a multicollector inductively coupled plasma mass spectrometer was used to measure the $^{238}U/^{235}U$ isotope ratio to discriminate between abiotic and biotic U transformation mechanisms[27]; and DNA-based (16S rRNA) microbial community analysis was performed in order to characterize its diversity and metabolic potential[32,33] (Supplementary Table 1). The experimental findings for this deposit may help shed light on other deposits of the same type and augment our current understanding of roll-front deposit formation.

## Results

**Non-crystalline $U^{(IV)}$ within roll-front deposits.** Sequential extractions (Fig. 1 and Supplementary Table 2) revealed that most of the U associated with roll-front sediments was bound to carbonate (~31%), organic carbon (~30%) and clay (~35%) fractions and <5% was associated with Fe/Mn oxides. We used bulk U $L_{III}$-edge extended X-ray absorption fine structure (EXAFS) spectroscopy to characterize the U species present in the roll-fronts by comparing the EXAFS spectra of the sediments with those of several $U^{(IV)}$ and $U^{(VI)}$ standards based on their likelihood of being present under similar experimental conditions. Table 1 clearly shows non-crystalline $U^{(IV)}$ as the dominant $U^{(IV)}$ species representing between 58% and 89% of the total U. Uraninite was another $U^{(IV)}$ species found within these sediments but was mostly a minor component (2-29%). A small fraction (7-17%) of U with a valence of +6 was found to be associated with metal oxides such as Fe (oxy)hydroxide. Bulk U-EXAFS and X-ray diffraction data did not exhibit any evidence of coffinite or other $U^{(IV)}$ ore minerals (Table 1, Supplementary Table 3). Rigorous examination of the data, including the determination of the local atomic coordination environments of U as obtained from the shell-by-shell fitting of U EXAFS spectra, revealed multiple U coordination environments with the majority of U being in the +4 oxidation state and bound to bidentate carbon (C1) and/or oxalate-like (C2 and $O_{dist}$) ligands (Fig. 2 and Table 2). While the model invokes contributions from U-U pair correlations indicative of $U^{(IV)}$ minerals such as uraninite, the

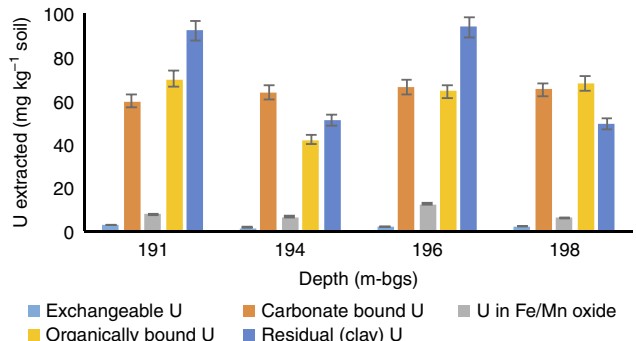

**Figure 1 | Sediment associated U fractions from sequential extractions.**
Bars indicate concentration of uranium (in mg kg$^{-1}$) within each fraction. Error bars indicate 1 s.d. from the mean of triplicate extractions. Total U concentrations (in mg kg$^{-1}$) for the given depths (m-bgs) as obtained from HF digestions are as follows: 191.0 m: 231.86 mg kg$^{-1}$; 194.0 m: 164.11 mg kg$^{-1}$; 196.0 m: 237.96 mg kg$^{-1}$; 198.0 m: 190.00 mg kg$^{-1}$.

**Table 1 | Bulk U EXAFS analysis.**

| Sample depth (m-bgs) | %U species from linear combination fitting of U EXAFS | | |
|---|---|---|---|
| | Biogenic UO$_2$ | Non-crystalline U$^{(IV)}$ | U$^{(VI)}$ associated with Fe(oxy)hydroxide-like phases |
| 191.0 | 29.1 | 58.4 | 12.5 |
| 193.8 | 11.2 | 88.8 | — |
| 194.0 | 15.7 | 68.8 | 15.5 |
| 194.4 | 2.2 | 81.5 | 16.3 |
| 196.0 | 4.1 | 88.4 | 7.4 |

EXAFS, extended X-ray absorption fine structure.
Percentage of U$^{(IV)}$ and U$^{(VI)}$ species within given sample depths (m-bgs) based on linear combination fitting of U EXAFS spectra using three phases; biogenic UO$_2$, non-crystalline U$^{(IV)}$ and U$^{(VI)}$ associated with Fe(oxy)hydroxide-like phases.

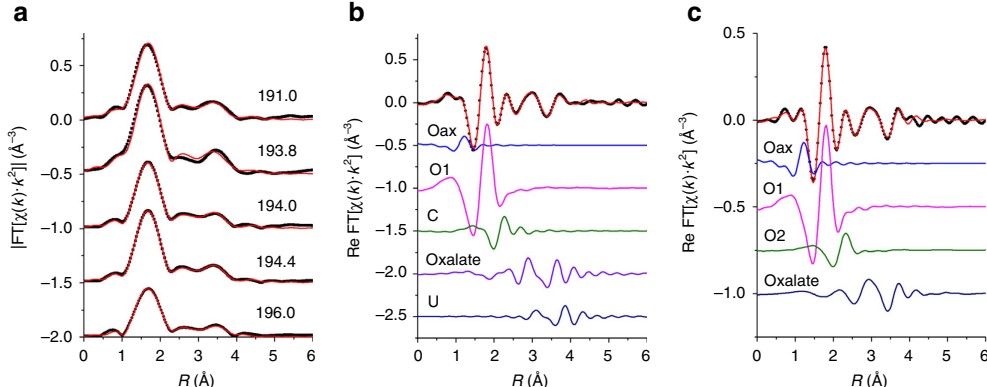

**Figure 2 | U L$_{III}$-edge EXAFS spectra and models.** (**a**) Magnitude of Fourier transform spectra are offset for clarity for following depths (from top to bottom in m-bgs): 191.0, 193.8, 194.0, 194.4 and 196.0. (**b,c**) Real part of Fourier transform of the sediments from depths 191.0 m (**b**) and 196 m (**c**). The components of the model are shown offset beneath the total model (i.e., O$_{ax}$, O1, C, Oxalate, U and O2). Contributions from O$_{ax}$: axial oxygen atoms from uranyl (U$^{(VI)}$); O1, O2: oxygen atoms bound to U$^{(VI)}$ and U$^{(IV)}$ at a longer distance than U-O$_{ax}$ C: carbon atoms from bidentate carbon group; Oxalate: oxalate-like ligand group; U: U neighbours in uraninite are shown in B. EXAFS spectra (symbols) and models (lines).

**Table 2 | U speciation from EXAFS analysis within given sample depths (m-bgs).**

| Sample depth (m-bgs) | Coordination number of U from shell-by-shell fitting of U EXAFS | | | | | |
|---|---|---|---|---|---|---|
| | O$_{ax}$ | O1 | O2 | C1 | C2 | U1 |
| 191.0 | 0.24 ± 0.06 | 9.7 ± 0.7 | — | 2.6 ± 0.8 | 7 ± 4 | 3 ± 4 |
| 193.8 | 0.14 ± 0.16 | 11.6 ± 1.1 | — | 2.9 ± 1.1 | 10 ± 6 | 5 ± 5 |
| 194.0 | 0.34 ± 0.05 | 8.2 ± 0.6 | — | 2.2 ± 0.6 | 7 ± 4 | 3 ± 4 |
| 194.4 | 0.34 ± 0.05 | 9.1 ± 0.7 | — | 2.6 ± 0.8 | 8 ± 4 | 3 ± 4 |
| 196.0 | 0.27 ± 0.04 | 4.9 ± 0.2 | 2.0 ± 0.3 | — | 8 ± 2 | — |

EXAFS, extended X-ray absorption fine structure.
O$_{ax}$: number of axial oxygen atoms from uranyl (U$^{(VI)}$). O1, O2: number of oxygen atoms bound to U$^{(VI)}$ and U$^{(IV)}$ at a longer distance than O$_{ax}$. C1: number of carbon atoms from bidentate carbon group. C2: number of carbon atoms from oxalate group. U1: Number of uranium neighbours in uraninite.

uncertainty in the U-U coordination number (CN) is as large or larger than the CN at all depths, indicating no significant contributions from uraninite (Table 2 and Fig. 2c). In other words, the same quality of fit between model and data can be obtained without contribution from the U–U pair correlation. In contrast, the uncertainty in the C2 CN, while large, is smaller than the CN itself, indicating that the model requires this signal to maintain the quality of fit. Further inspection comparing the Fourier transform of the EXAFS spectrum for the specific sample (191.0 m-bgs) exhibiting the highest contribution of UO$_2$ as determined by linear combination fitting (LCF) (29%) with that of crystalline uraninite illustrates the large difference in their spectra (Supplementary Fig. 3). Thus, we conclude that none of the samples contain measurable crystalline UO$_2$, further strengthening our finding that U in these natural sediments is

predominantly non-crystalline U$^{(IV)}$. The EXAFS modelling results shown here are similar to those found previously in non-ore sediments suspected of biological reduction of U$^{(VI)}$. Incomplete U$^{(VI)}$ reduction resulting in U coordinated to oxygen with a residual axial oxygen contribution with a CN < 2 is common[27]. The presence of bidentate carbon (C1) ligands at 2.9 Å is also common[16,17,20,23]. Slightly less common is the presence of oxalate-like ligands with U-C2 and U-O$_{dist}$ interatomic distances of 3.4 and 4.4 Å. Both the sequential extraction and the shell-by-shell EXAFS data suggest that a large fraction of U in the solid phase is bound to the organic sediment fraction (representing 3.6-4.9 % of the dry sediment w/w, Supplementary Table 4), represented by oxalate-like functional groups. Previous characterization of post-mining ore zone samples also showed association between U and carbonaceous

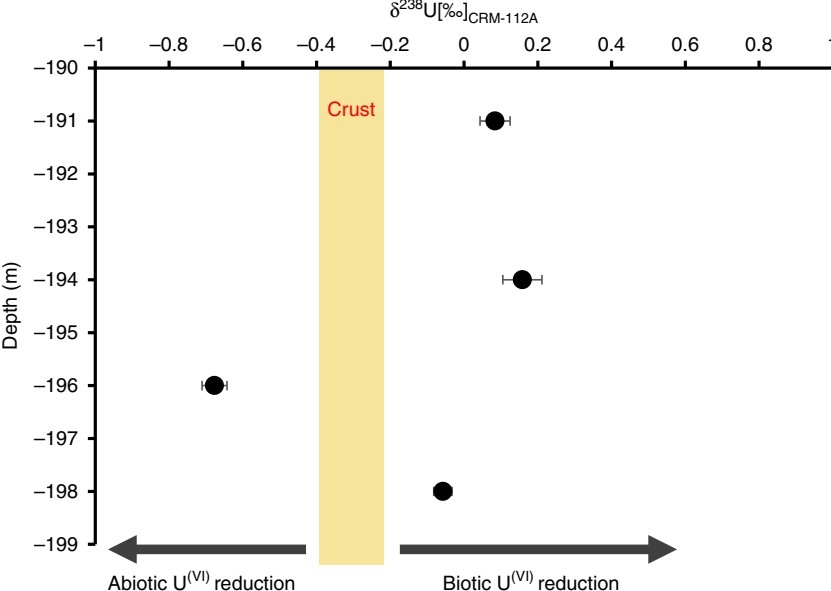

**Figure 3 | δ²³⁸U values for U roll-front deposits at four depths.** 191.0, 194.0, 196.0 and 198.0. Error bars indicate two s.d. from the mean of triplicate extractions. The yellow box at $\delta^{238}U = -0.2$ to $-0.4‰$ indicates the average U isotope signature of the Earth's crust and is often considered as the reference to delineate between abiotic vs biotic U reduction[41,42]. CRM-112A was used as the standard. $\delta^{238}U$ values in (‰) and depth in (m-bgs).

material in the deposit[24,33–35]. Thus, the results presented show direct evidence for the presence of non-crystalline U(IV) within undisturbed roll-front deposits, a revision to the established paradigm[29,36].

**U isotopic signature**. In order to investigate the reduction mechanism (abiotic vs. biotic) through which the non-crystalline U(IV) species are formed and subsequently stabilized within the roll-front deposits, we adopted a novel tool based on the fractionation of the two primordial U isotopes: $^{238}U$ and $^{235}U$, reported as $\delta^{238}U(‰) = [(^{238}U/^{235}U)_{sample}/(^{238}U/^{235}U)_{standard} - 1] \times 1000$ (refs 22,25,37,38) (Fig. 3 and Supplementary Table 5). Stylo et al.[27] found that biotic reduction generates heavy U isotope signatures in the solid phase, consistent with previous laboratory and theoretical findings[39,40], while abiotic reduction results either in depletion of $^{238}U$ in the reduction product or in no isotopic fractionation; the latter observation is inconsistent with theoretical considerations assuming equilibrium isotope fractionation between U(IV) and U(VI), but is consistently observed in abiotic reduction experiments[22,27,37]. Thus, sediment $\delta^{238}U$ can serve as a signature for past or present biological activity in sedimentary rocks. In three out of four samples, the sandstone of the roll-front deposit investigated here displays higher $\delta^{238}U$ than those typically observed for the continental crust ($\delta^{238}U \sim -0.3‰$)[41–44] consistent with a predominantly biotic reduction mechanism (Fig. 3 and Supplementary Table 5). Combining the findings, the isotopic measurements and EXAFS spectroscopy strongly support biotic reduction of U(VI) as the dominant reduction pathway for the generation of non-crystalline U(IV) within roll-front deposits. Only one of the investigated samples (196 m-bgs) displayed a $\delta^{238}U$ lower than that typical for the continental crust. As U isotope fractionation during oxidative U mobilization is assumed to be negligible[45], this might be the result of either U reduction from a fluid depleted in $^{238}U$ by previous biotic U reduction or U reduction with abiotic reducing agents, indicating a redox dynamic system with simultaneous biotic and abiotic reduction.

**Microbial communities present within ore deposits**. Microbial community analysis showed the presence of bacteria closely related to known U-reducing organisms, including organisms in the genera *Pseudomonas*, *Clostridium* and *Geobacter* (Supplementary Table 6)[46,47]. Although the actively reducing population, if any, cannot be conclusively identified from these results, the presence of potential U(VI)-reducing organisms is in accordance with isotopic and spectroscopic evidence of biogenic U(VI). The presence of both U(VI) and reduced minerals (for example, U(IV) and pyrite; Supplementary Table 3)[30] suggests that current geochemical conditions maybe similar to the original depositional environment. If this is true, then the presence of U-reducing organisms is consistent with the isotopic evidence that non-crystalline U(IV) is biogenic in origin.

Our findings establish that non-crystalline U(IV) is the major form of U(IV) in the ore and that it is likely formed via biotic U(VI) reduction during ore genesis. Bioreduction of U(VI) can occur through several mechanisms, depending on the microbial species involved. *Pseudomonas, Geobacter* and *Clostridium* species are capable of reducing U(VI) enzymatically to form non-crystalline U(IV) (refs 47–49) Our field data are in agreement with recent laboratory-based studies showing that biofilms of *Geobacter sulfurreducens* were able to immobilize and reductively precipitate U(VI) into non-crystalline U(IV) phase via bonding to carbon ligands[47–49]. At one depth (196.0 m-bgs), our isotope fractionation data indicated that abiotic U reduction may have been the dominant reduction mechanism, but the major product is still non-crystalline U(IV). The possibility of abiotic reduction has also been documented by WoldeGabriel et al. who observed U(IV) on lignite and pyrite surfaces in sediments mined by an alkaline ISR process at the Smith Ranch Highland mine[24]. On the basis of our spectroscopy and isotope data, it is observed that the chemical nature of the sediment at 196.0 m-bgs was distinct from the others. A possible explanation could be that U(VI) was reduced chemogenically by solid-phase ferrous iron since we find a higher fraction of U bound to Fe oxides at this particular depth (Fig. 1). However, previous findings have shown that abiotic U(VI) reduction mediated by Fe(II) within Fe bearing oxides leads to formation of U(IV) nanoprecipitates[50] but abiotic reduction of

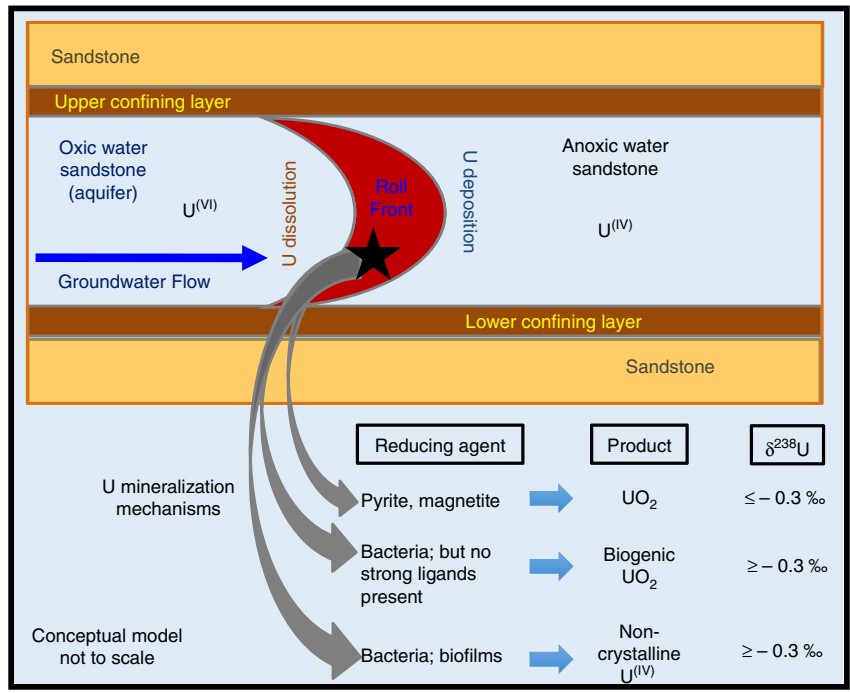

**Figure 4 | Conceptual model of uranium roll-front deposit formation.** Precursors and conditions favouring the formation of uraninite, biogenic uraninite and non-crystalline U$^{(IV)}$ are illustrated.

U$^{(VI)}$ in association with biofilms generates non-crystalline U$^{(IV)}$ (refs 20,22). Another possibility is reduction by reduced organic functional groups resulting in non-crystalline U$^{(IV)}$ formation, because that mechanism was shown to result in no significant U-fractionation[25]. These results show that roll-front deposits are heterogeneous and contain microenvironments where either abiotic or biotic U reduction can occur. Overall, this study clearly shows that biotic reduction is important and maybe the dominant U reduction mechanism in this ore body. Interestingly, the long-term stability of non-crystalline U$^{(IV)}$ is not known, but its presence in the ore body suggests that it maybe more stable in reducing environments than previously suggested.

**Implications for uranium ore genesis and U mining**. The global production of U was 56,217 tons in 2014, half of which was mined by ISR of roll-front deposits[51]. Thus, predictive models of profitability based on extractable U within roll-fronts and ISR site restoration strategies need to include non-crystalline U$^{(IV)}$ species because their solubility and redox properties are likely to differ from crystalline U forms[50]. There is significant uncertainty with respect to the thermodynamic and kinetic properties of non-crystalline U$^{(IV)}$. It was reported that this form of U is not only a short-lived intermediate that undergoes transformation to uraninite[52], but also that it can be stable for multiple years under certain conditions[16,53]. The prevalence of non-crystalline U$^{(IV)}$ as a reduced uranium species suggests the long-term persistence of this species in this environment. Nonetheless, future studies are needed for detailed physico-chemical characterization of non-crystalline U$^{(IV)}$ in other sandstone-hosted U ore deposits[54–58] in order to quantify the global significance of this form of U and to improve our current understanding of its mobility, toxicity and environmental impact.

On the basis of our experimental findings, we present a schematic diagram highlighting the U reduction mechanisms leading to the formation of primarily biogenic non-crystalline U$^{(IV)}$ (Fig. 4). The prevalence of non-crystalline U$^{(IV)}$ has not previously been recognized in roll-front U deposits, and has

implications for models of ore genesis and impacts future human health-risk assessment, post-mining site restoration and prediction of economically recoverable U in similar roll-front deposits. Our findings, if confirmed in other roll-front deposits, may require a paradigm shift in current post-mining restoration models as well. While abiotically produced uraninite and coffinite were thought to be the major components of uranium ores, and are the basis for estimations of economically recoverable uranium from ores, this study shows that U maybe trapped primarily via biological reduction in roll-front deposits, resulting in high fractions of non-crystalline U$^{(IV)}$ bound to C either from organic functional groups or inorganic carbonate. Thus, non-crystalline U$^{(IV)}$ formed by direct enzymatic U reduction in ore formation maybe more important than previously thought. Many low-temperature ores display similar isotopic fractionation[59,60] as found here suggesting an under-recognized importance of biological processes in roll-front ore formation.

## Methods

**Study site description.** The experimental site used for these studies is located in the southern part of the Powder River Basin in Converse County, WY, USA (Supplementary Fig. 1a). The Smith Ranch-Highland site is the largest ISR U mine in the US, operated by Cameco Resources Inc., with an average grade of 0.10% as U$_3$O$_8$, and producing over 43.3 million pounds of yellow cake per year (https://www.cameco.com/about). ISR mining is a solution mining technique, which is used to extract U from low-grade ore-deposits by introducing oxygen and/or H$_2$O$_2$- and CO$_2$-fortified native groundwater via injection wells. The oxygen in the water oxidizes the U$^{(IV)}$ into its soluble U$^{(VI)}$ form. Bicarbonate promotes the formation of highly stable and mobile U$^{(VI)}$(hydroxy)carbonate and calcium-uranyl-carbonato ions. The U-bearing water is then pumped to the surface, where the uranium is extracted through an ion-exchange process. Consequently there is little surface disturbance and no tailings or waste rock generated during ISR U mining operations as compared to conventional mining. The sandstone units, interbedded shales, and high-carbonate zones may contain between 1 and 20 mineral fronts. The Paleocene Fort Union Formation is over 305 m thick and only the upper 213 m contains the arkosic sandstone units with associated U mineralization. The ore deposit that is the focus of this study is located in the saturated zone of the confined aquifer and is ∼6 m thick[3].

**Sediment collection and preservation.** Soil/sediment cores were collected in spring of 2013 from roll-front deposits of an unmined site at Smith-Ranch

Highland (Supplementary Fig. 1b). The logs and ore-grade data for the mine unit from where the core was collected are provided within the Supplementary Data 1. A map showing the location of the core in Mine Unit 3 extension is shown in Supplementary Fig. 2. The bore–hole from which the core was taken was cased and completed, and the log was done after casing. The 'ore-grade' printout (please refer to Supplementary Data 1) shows the estimated weight percent U at various depths and indicates that one zone reaches 0.2% U. The map (Supplementary Fig. 2) shows the production mine unit with MOW 3-1 (where the core was taken) and monitoring wells near the ore body. It is to be noted that none of these wells had yet started operating when the core was taken and represented complete native conditions. Sections (0.3 m long) of the sediment cores were vacuum-sealed in freezer bags as the core was extruded in the field to preserve redox conditions and brought back to the laboratory on dry ice. Once back in the laboratory, the sediment cores were split open and homogenized inside an anaerobic glove box (with 3-4% $H_2$ and a balance of $N_2$) to prevent oxidation of U. Initial sample preparation included careful removal of drilling mud from the outer layer of the sediments using a chisel. Approximately 5 g of sample was homogenized under sterile conditions in the anaerobic chamber immediately upon thawing of the frozen drill core, placed in a sterile, sealed tube, removed from the anaerobic chamber and frozen for DNA analysis. For geochemical experiments, the sediments were lightly disaggregated with a mortar and pestle and allowed to dry before being passed through a 2 mm sieve (allowing the majority of the sediment to be characterized including 'very coarse sand'). The homogenized sediments were stored in sealed glass serum bottles until geochemical and isotope experiments.

**Geochemical methods.** Sequential extraction procedures for U were performed based upon a modified method from Tessier et al.[31] as described in Salome et al.[30] to extract the soluble and exchangeable, carbonate-bound, Mn- and Fe oxide-associated, organic bound and residual U. The following procedure was performed sequentially: (1) 4 ml of 1.0 M $MgCl_2$ (pH 7.0) was added to ∼0.5 g sediment and agitated at 20 °C for 1 h to extract the exchangeable U fraction; (2) 4 ml of 1.0 M sodium acetate (adjusted to pH 5.0 with 1.0 M HCl) was added and agitated at 20 °C for 5 h (ref. 61) to extract U associated with the carbonate fraction; (3) 10 ml of 0.04 M $NH_2OH \cdot HCl$ in 25% (v/v) acetic acid was added and agitated at 96 °C for 6 h to extract U associated with the Fe/Mn-oxides fraction; (4) 1.5 ml of 0.02 M $HNO_3$ and 2.5 ml of 30% $H_2O_2$ (pH 2.0) were added and agitated at 96 °C for 2 h, a second 1.5 ml aliquot of 30% $H_2O_2$ (pH 2.0) was added and agitated at 96 °C for 3 h, and a third 5 ml aliquot of 2.5 M $NH_4OAc$ in 20% (v/v) $HNO_3$ was added and agitated at 20 °C for 1 h to extract U associated with the organic fraction; and (5) 5 ml of 15.8 M $HNO_3$ was added and maintained at 85 °C for 3 h to extract the residual fractions[62]. After each extraction step, samples were centrifuged (1,380 g for 10 min), and supernatants were filtered (0.2 μm, PES Puradisc Whatman) for analysis by inductively coupled plasma mass spectrometry.

Samples were prepared for X-ray diffraction using a modified method based on Eberl[63]. One gram of homogenized sample (<2 mm fraction) was mixed with 20% corundum and ground in a McCrone micronizing mill with 4 ml ethanol for 5 min, generating particle sizes on the order of 10–30 μm. After drying at 60 °C, the mixture was transferred to a plastic scintillation vial with three acrylic balls (∼1 cm in diameter) along with 200–800 μl Vertrel solution (Dupont) and shaken for 10 min. The powder was passed through a 250 μm sieve to break up the larger aggregates and loaded onto an X-ray diffraction sample holder. Samples were analysed using a Siemens D500 X-ray diffractometer from 5 to 65° 2θ using Cu KαX-ray radiation, with a step size of 0.02° and a dwell time of 2 s per step. Quantitative mineralogy was calculated using the USGS software, RockJock[63], which fits X-ray diffraction intensities of individual mineral standards to the measured diffraction pattern.

XAS was used to determine the bulk U oxidation state and molecular coordination environment in the sediments. U $L_{III}$-edge extended X-ray absorption fine structure (EXAFS) spectrum refers to the oscillatory part of the U-XAS spectrum which occurs above the U absorption edge (17,166 eV). The signal in this region yields information on the local chemical coordination environment of U (CN, identity of ligand atom, and distance between U and ligand atom) in an unknown sample (sediments, in our case). Theoretical calculations of these signals can be used to model the U EXAFS spectrum by comparing signals from known U containing model compounds (U-C, U-O, U-P) in the first and second coordination shells. This fitting technique is widely used in natural samples where U can be present in myriad forms (adsorbed, co-precipitated and precipitated). Bulk U $L_{III}$-edge (17,166 eV) EXAFS data were collected for sediments with sufficient U concentrations. The data were collected at beamlines 4-1 and 11-2 at the Stanford Synchrotron Radiation Lightsource in Menlo Park, CA under ring-operating conditions of 3 GeV with a current of 450 mA. All sample preparation was conducted in an anaerobic glove bag containing 3-4% $H_2$ and a balance of $N_2$. Samples were packed in teflon holders and sealed with Kapton tape to preserve the oxidation state of U. During analysis, samples were mounted in a cryostat maintained at 77 K using liquid nitrogen to prevent beam damage. A double-crystal Si (220) monochromator was used for energy selection, detuned 15–30% to reject higher harmonic intensities, and was calibrated using Y foil as the internal standard. The Y foil was also used as an internal calibrant by simultaneously measuring the transmission spectra of the foil and each sample scan. EXAFS oscillations were subtracted by fitting a smoothly varying function (spline) to remove contributions below 1.4 Å, which may result in non-physical pair

correlations, using the SixPACK[64] and Athena analysis packages[65]. EXAFS data were normalized, background subtracted and analysed using the SIXPACK and Horae program packages. LCF of the normalized spectra was performed using the Athena program. LCF of spectra was performed in $k^3$-weighted k-space between $k = 3$ and 10.2, using three end members: $U^{(VI)}$ adsorbed to ferrihydrite as a $U^{(VI)}$ reference; biogenic uraninite defined as short-range ordered $U^{(IV)}$ associated with biomass and formed via microbial reduction; and non-crystalline $U^{(IV)}$: nano-particulate short-range ordered uraninite lacking the structure of crystalline inorganic uraninite and often associated with biofilms and/or inorganic metal oxides. These end members were chosen based on their likelihood to be present under the experimental conditions. Compounds were only included in the fit if the contribution was a fraction >0.05.

The UWXAFS package[66] was used to perform the modelling of the U-EXAFS spectra. The final theoretical model was built using FEFF7 (ref. 65) based on crystallographic structures such as $U^{(IV)}$ dioxalatehexahydrate and $U^{(IV)}$ acetate. The error analysis and the goodness-of-fit parameters were calculated by the fitting routine FEFFIT[66]. The model consists of scattering paths of the photoelectron from the first few neighbouring shells of atoms about the U atoms in the sample. Initially the spectra were tested for O, C, Si, Al, Fe, P and U neighbours including several different bonding distances and geometries. Only the best-fit models with reasonable fitting results are presented. The data were processed by using $\Delta k = 1.3$ to 8.0–8.5 Å$^{-1}$ and $\Delta R = 1.0$–4.2 Å with a Hanning window of sill width 1.0 Å$^{-1}$. The modelled data and Fourier transform ranges resulted in 16-14 degrees of freedom in the fit per data set. Four data sets from sediment depths of 191.0, 193.8, 194.0 and 194.4 m were simultaneously optimized. The model contained 12 shared parameters and five independent CNs. Each data set was refined against an average of 8 (=12/4+5) parameters. The U EXAFS spectrum from sample depth of 196.0 m showed a much smaller first shell oxygen signal requiring an independent model (Fig. 2a). This model contained 11 parameters refined against 16 degrees of freedom in the data set. The resulting EXAFS CNs, $\Delta$Rs and $\sigma^2$ values are given in Table 2 and Supplementary Tables 7 and 8. Figure 2b,c show the EXAFS data and model with the individual contributions offset beneath. Enlarged ball-n-stick figures representing the $U^{(IV)}$ species are shown in Supplementary Fig. 4. Each contribution to the model is described in Supplementary Tables 7 and 8. The models contains a small amount of axial oxygen atoms (<0.5) indicating that most of the U is $U^{(IV)}$. The percentage of the $U^{(VI)}$ is estimated based on the U-O$_{ax}$ CN. The next shell in the model is from longer distance oxygen (O1 and possible O2) atoms. $U^{(VI)}$ and $U^{(IV)}$ are usually coordinated by 6 and 8–10 oxygen atoms, respectively, at a distance of 2.1–2.6 Å. Some $U^{(IV)}$ compounds have U-O1 distances as long as 2.8 Å as found for sample depth 196.0 m (Fig. 2c). The best-fit values for the number of oxygen atoms (O1 and O2) range from 7-11.6 ± 1 (Table 2) as expected for $U^{(IV)}$. There is a slight trend of more oxygen atoms in samples with slightly less $U^{(VI)}$ as expected. The average distance for U-O1 signal is 2.37-2.36 ± 0.01 Å and is in the expected range. The $\sigma^2$ value is large (0.016–0.020 ± 0.002 Å$^2$) indicating a good deal of disorder (Supplementary Table 8). The next shell contains carbon atoms at 2.81 ± 0.01 Å for samples depths 191.0, 193.8, 194.0 and 194.4 m. This is a typical distance for bidentate carbon ligand from either carbonate or acetate. Multiple scattering from distance C or O from this group was not required, so it is uncertain which type of ligand this bidentate C group represents. The average number of C atoms is 2–3 ± 1 and is consistent with an average of at least one to two bidentate carbon containing groups as depicted in Supplementary Fig. 4. The disorder in this signal ($\sigma^2 = 0.001 ± 0.007$ Å$^2$) is small but with large uncertainty (Supplementary Table 7). The next signal comes from carbon and distant oxygen atoms consistent with oxalate-like groups. The U-C2 and U-O$_{distal}$ distances are 3.4 and 4.4 Å, respectively as found in $U^{(IV)}$ dioxalatehexahydrate. The C2 CN ranges from 7 to 10 ± 4–6 atoms. The large uncertainty in the CN is due to overlap with possible U-U signal also in this range. On the lower end, our model is consistent with two to three oxalate-like groups, each contributing 2 C atoms as shown in Fig. 2b,c and Supplementary Fig. 4. The final signal included in the model is from U at 3.8 Å as found in uraninite. The CN for this U signal is small (3–5 ± 4–5) compared to uraninite[13] and the uncertainty is larger than the measured value such that it is not definitive measurement of uraninite. If there is uraninite, it is not a major component in these samples as illustrated in Supplementary Fig. 3 by directly comparing the measured EXAFS spectrum of uraninite with that of the sediment spectra.

EXAFS is a unique tool for determining the U speciation within the sediments because the signal originates only from the U atoms within the samples with no competing signal from other crystalline or non-crystalline components. There is a minimum amount of U needed in the sample to obtain a measureable signal. This limit depends on the details of the measurement for which we estimate that limit to be ∼50 mg kg$^{-1}$. There is also a limit to the resolution of EXAFS to determine minor phases within a sample. In general this limit is ∼5–10% depending on the specific nature of the phase. The EXAFS signal is a sum of the signals from all the U atoms within the sample. In general a signal from 0.5 atoms can be detected. For example, if the sample contains 50% of $UO_2$, and each U atom in this phase is coordinated by 12 U atoms, then the EXAFS spectrum would contain a signal from 6 U atoms which is easily detected. Less than 5% of $UO_2$ would give a signal of 0.6 U atoms, which would be at the edge of our detection limit.

**U isotopic measurements.** Uranium was purified by ion-exchange chromato-graphy following the method by Weyer et al.[38] Before the U-extraction ∼50 mg of

the sediments were ashed at 600 °C for 12 h to oxidize the organic material. Samples were digested in several steps: (1) with 2 ml 6 M HCl at 130 °C for 24 h; (2) with 4 ml of a mixture of 15 M HNO$_3$ and 2 ml 24 M HF (1:1) at 130 °C for 24 h; (3) once more with 2 ml 6 M HCl at 130 °C for 24 h; (4) with 2 ml 15 M HNO$_3$ at 130 °C for 24 h and finally with 3 ml of 6 M HCl and 1 ml 15 M HNO$_3$ (3:1 aqua regia) at 130 °C for 24h. The latter step resulted in a clear solution in all cases. For the U ion-exchange chromatography all samples were dissolved in 1 ml 3 M HNO$_3$ and spiked with a $^{233}$U–$^{236}$U isotope tracer (IRMM 3636-A) in order to correct for potential isotope fractionation during U separation and instrumental mass discrimination during measurements[31]. The $^{238}$U/$^{235}$U isotopic composition was measured with a Thermo-Neptune multicollector inductively coupled plasma source mass spectrometer at the Institute for Mineralogy at Leibniz Universität Hannover. For sample introduction, an ESI Apex nebulizer (without membrane) was coupled to the desolvation unit of a CetacAridus. The measuring protocol followed a standard sample bracketing method (that is, every two samples were bracketed by a CRM-112A standard). The results for all sample analyses are reported in the delta notation relative to the CRM-112A standard:

$$\delta^{238}\text{U} = \left[ \frac{(^{238}\text{U}/^{235}\text{U})_{\text{sample}}}{(^{238}\text{U}/^{235}\text{U})_{\text{CRM}-112\text{A}}} - 1 \right] \times 1000 \qquad (1)$$

Three replicate analyses were performed for each sample. The precision is reported as two s.d.'s of the replicate analyses for each sample, which is typically ≤ 0.1‰. Analytical quality, that is, the accuracy of our analytical protocol, has been frequently tested by replicate analyses of the U standards REIMEP 18-A and IRMM-184 relative to CRM-112A. The results for both standards agreed within those previously reported in the literature[67]. The sample concentration of the sediments and water samples were measured with inductively coupled plasma optical emission spectroscopy (ICP-OES) (Vista Pro, VARIAN).

**Microbial community analysis.** DNA was extracted using a PowerSoil extraction kit (MoBio Laboratories) under sterile conditions in a laminar flow hood. Although the DNA recovery was low (<1 ng ul$^{-1}$), amplification of the V1-V3 regions of the 16S rRNA gene (28F-519R primers) was successful for samples reported here. Amplification and sequencing was performed at Research and Testing Laboratories (Lubbock, TX) on a Roche 454-FLX/FLX+ platform with ~10,000 sequences returned per sample. The rarefaction curve shows that the sequencing depth was appropriate to represent the community (Supplementary Fig. 5). Sequences were denoised, aligned and analysed with QIIME[32] using the SILVA database[68]. Operational taxonomic units were identified at 97% sequence identity and grouped taxonomically at the genus level (Supplementary Table 6).

**Data availability.** All relevant data are available from the authors or within the Supplementary Information files.

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

## Acknowledgements

This project was funded by the University of Wyoming, School of Energy Resources, which administered funds appropriated by the Wyoming State Legislature for research activities related to uranium ISR in Wyoming. Additional support was provided by the Toxic Substance Hydrology Program at the US Geological Survey. We thank James Clay of Cameco Resources for providing access to the Smith Ranch-Highland ISR site and for logistical and technical support of the field sampling, and to Richard Wanty (USGS) for review comments. Use of the Stanford Synchrotron Radiation Lightsource, SLAC National Accelerator Laboratory, is supported by the US Department of Energy, Office of Science, Office of Basic Energy Sciences under Contract No. DE-AC02-76SF00515. Any use of trade, firm or product names was for descriptive purposes only and does not imply endorsement by the US Government.

## Author contributions

A.B., K.M.C., R.B.-L. and T.B. designed research. A.B., K.M.C., Y.R. and T.B. performed the research. S.W. contributed new analytic tools. A.B., K.M.C., S.K., S.W. R.B.-L and T.B. analysed the data. A.B., K.M.C., S.K., S.W., R.B.-L. and T.B. wrote the paper.

## Additional information

**Competing interests:** The authors declare no competing financial interests.

