## [Peer Review File · Nature Communications]

Reviewers' comments:

Reviewer #1 (Remarks to the Author):

The manuscript is generally well written and offers an interesting set of data to support the authors hypothesis that U(IV) ore deposits can be laid down by microbial action. This has been proposed previously in the extensive literature on uranium reduction (over the last 2 decades), but this is the first manuscript that I have seen to use the combination of EXAFS and isotopic analyses to test the hypothesis. The work has merit, but there are issues. I have the following comments:

L72 What is an “in-situ recovery U mine”? Should be defined for non-specialist.

L77 “call into question the prevailing model for uranium ore formation and the U speciation therein and warrant the re-evaluation of the paradigm in order to optimize ore exploration and exploitation, as well as mine restoration.” This is grand claim, and one wonders whether this is supported by analysis of the restricted number of samples (which is not unequivocal; see below)

L109 “Tab. 1a” in the main body of the text, write out in full. Also fitting needs the “end-member” spectra identified in the caption.

L111 “Biogenic uraninite” how is the uraninite identified as biogenic in origin?

L114 “There was no evidence of coffinite or other U(IV) ore minerals present.” Therefore, the samples analyzed would seem to be significantly different to samples characterized previously as “Based on X-ray diffraction (XRD), a previous study identified pitchblende, uraninite and coffinite as the main U-bearing minerals in the ore³⁰.” One wonders how representative the samples in this study were of those studied previously, and how heterogeneous the mineralogy of the site is (which clearly impacts on extrapolation; see L77 comment).

L118 “While the model invokes contributions from U-U pair correlations indicative of U(IV) minerals such as uraninite” is discounted, but could have been strengthened by other mineralogical analyses e.g. HRTEM (which would also help strengthen the overall mineralogical analysis of the sediments). In my opinion, too much hangs on the EXAFS analyses, which is a little difficult to follow and seems to have been presented rather selectively.

L157, the authors imply a high abundance of uranium-reducing organisms in the samples (“0.1-31%”) but this is not really true of those that are known to reduce U(VI). The authors note these include “Pseudomonas, Clostridium, and Geobacter (Supplementary Tab. S4); other core samples from the same mine site also contained Desulfovibrio and Shewanella.” Pseudomonads are not normally associated with U(VI) reduction (and the reference cited to support this is on U(VI) reduction by Clostridium species). As the 31% abundance of the “uranium-reducing organisms” corresponds to the Pseudomonads in the 194 m bgs sample, it is misleading to note that “0.1-31%” of the sequence correspond to uranium reducing organisms. Shewanella species, which feature in the discussion (due to the work on non-

crystalline U(IV) formation by this model organism) are not shown in table S4 so are presumably at extremely low abundance (and probably not relevant). Interestingly, Geothrix is one metal-reducing organism which was detected at relatively high levels in the samples (at 18.6% in the 188 m bgs sample) but it is not discussed at all. One could argue that it could potentially play a role in uranium reduction, although little work has been done on this organism in this context but it has been associated with sediments undergoing U(VI) bioreduction previously in other studies. In general however, it is difficult to describe the microbial communities detected as those expected in a zone of substantial metal/uranium bioreduction as implied in the manuscript (especially for sample 194 which seems to be the only sample that has been analysed for U speciation, isotopes and microbial composition).

L 188, The authors note that “These results indicate that roll-front deposits are heterogeneous and contain microenvironments where either abiotic or biotic U reduction can occur. Nonetheless, this study clearly shows that biotic reduction is important and maybe the dominant U reduction mechanism in this ore body. The long-term stability of non-crystalline U(IV) is not known, but these data suggest that it may be more stable under reducing environmental conditions than previously suggested.” Although I agree about the heterogeneity, the comments about the dominant mechanism in this ore body do not seem adequately supported, and the comments about long-term stability seem overly speculative based on this study; just as likely is that the authors could have sampled a very dynamic redox system with recently reduced/precipitated U(IV)?

L 264 “Bulk U LIII-edge (17166 eV) extended x-ray absorption fine structure (EXAFS) data were collected for sediments with sufficient U concentrations.” What is limit of sensitivity for these analyses, and how does this relate to ore concentrations that are exploited in the body

Looked at holistically, the data are confusing and not always consistent with the hypothesis proposed. For example, the 196 m sample, is described as 88% non-crystalline U(IV) from the EXAFS, but from the $\delta^{238}\text{U}$ values it is identified as “abiotic” in origin. Also, it is concerning that the clear peaks expected for U-U in the EXAFS spectra are discounted; they seem quite pronounced in Figure 3 (at about 3.5 Å). This feature seems to be present even in the sample from 194.4 m-bgs, but in the legend in S4 it is noted as absent (although the plots are not shown in S4). For S-4 it would be useful to see the plots from all the samples. I would question that at L120 “While the model... invalidating...”, if you look at the C2 fits, the errors are large (typically >50 %) and thus there is perhaps a more reasonable view that there is significant uncertainty in this area within the EXAFS due to the complexity of the fits and really whether it is C2 or U-U is unclear although the C2 looks perhaps slightly better fitting. I am not sure you can be as strong as saying U-U is invalidated. I wonder what happens if you exclude the C2 and introduce the U – U as the main fit – does this give realistic values?

Reviewer #2 (Remarks to the Author):

Review of "Biogenic non-crystalline U(IV) revealed as major component in uranium ore: Implications for U ore genesis and mining"

The submitted manuscript by Bhattacharyya et al. investigates the role of bacterially-mediated reduction of U(VI) to U(IV), producing non-crystalline bound to C from either organic functional groups and/or inorganic carbonate in the Wyoming low temperature U roll-front ore deposits. The microbial community was assessed using DNA-based techniques, and the uranium species was investigated by using sequential leaching and EXAFS. A handful ($n=4$) $\delta^{238/235}\text{U}$ isotope results were also presented. The authors suggest that biogenic processes are a more important process during uranium ore genesis than previously understood, which has potential implications in mining applications, both for U recovery and bioremediation.

Whilst I feel that this manuscript is well written and topical, and the data appear robust, the role of bacterially-mediated reduction of U(VI) to U(IV) in low temperature U ore deposits has been identified in numerous other studies (many of which have not been cited in this manuscript), and U biomineralisation is well established as an important mechanism through which the low temperature uranium ore deposits form. An earlier study by Fletcher et al. (2010) (cited) presented EXAFS data from U reduction experiments using live *Desulfitobacterium* cultures, and showed that the U(IV) product was a phase or mineral composed of mononuclear U(IV) atoms closely surrounded by light element shells, which is likely the result of inner-sphere bonding between U(IV) and C/N/O- or P/S-containing ligands, such as carbonate or phosphate. This previously published finding is similar to that of this study. Thus, my primary concern with this manuscript is the lack of a particularly original and novel story.

A second concern is that many of the geochemical techniques were not applied to the same samples e.g. the bacterial assays were not conducted on the same samples as the isotopic measurements/EXAFS samples - so it is not possible to unequivocally say that the isotopic signatures are a result of biological/abiotic reduction without knowing the exact bacteriological species present. This study would benefit in general from additional isotopic measurements, particularly on samples that where it is suggested that abiotic mineralisation is the dominant U reduction mechanism. Whilst this might not be possible, measurement of the isotopic signatures for each of the sequential extractions would be interesting, particularly to see if the fractions associated with Fe/Mn oxides give a ~ 0 to 0.2 ‰ isotopic shift as observed by Brennecka, G. A., Wasylenki, L. E., Bargar, J. R., Weyer, S., & Anbar, A. D. (2011). Uranium Isotope Fractionation during Adsorption to Mn-Oxyhydroxides. *Environmental Science & Technology*, 45(4), 1370-1375.

It also would be useful to know where in the deposit the samples come from - e.g. high grade/low grade/unmineralised, up-gradient/down-gradient of the roll-front, within the limb/nose etc and any U concentrations of the samples.

Some examples of relevant biomineralisation literature:

Reynolds, R. L., Goldhaber, M. B., & Carpenter, D. J. (1982). Biogenic and nonbiogenic ore-forming processes in the south Texas uranium district; evidence from the Panna Maria deposit. *Economic Geology*, 77(3), 541-556.

Milodowski, A. E., West, J. M., Pearce, J. M., Hyslop, E. K., Basham, I. R., & Hooker, P. J. (1990). Uranium-mineralized micro-organisms associated with uraniferous hydrocarbons in southwest Scotland. *Nature*, 347(6292), 465-467.

Min, M., Xu, H., Chen, J., & Fayek, M. (2005). Evidence of uranium biomineralization in sandstone-hosted roll-front uranium deposits, northwestern China. *Ore Geology Reviews*, 26(3-4), 198-206.

Bargar, J.R., Bernier-Latmani, R., Giammar, D.E., & Tebo, B.M., (2008). Biogenic Uraninite Nanoparticles and Their Importance for Uranium Remediation. *Elements*, 4(6), 407-412.

Chen, Z., Cheng, Y., Pan, D., Wu, Z., Li, B., Pan, X., ... Guan, X. (2012). Diversity of Microbial Community in Shihongtan Sandstone-Type Uranium Deposits, Xinjiang, China. *Geomicrobiology Journal*, 29(3), 255-263.

Islam, E., & Sar, P. (2016). Diversity, metal resistance and uranium sequestration abilities of bacteria from uranium ore deposit in deep earth stratum. *Ecotoxicology and Environmental Safety*, 127, 12-21.

Kumar R., Nongkhaw M., Acharya C., Joshi S.R. (2013) Uranium (U)-tolerant bacterial diversity from U ore deposits of Domiasiat in North-East India and their prospective utilisation in bioremediation. *Microbes Environ*, 28(1), 33-34.

Sarma, B., Acharya, C., & Joshi, S. R. (2016). Characterization of Metal Tolerant *Serratia* spp. Isolates from Sediments of Uranium Ore Deposit of Domiasiat in Northeast India. *Proceedings of the National Academy of Sciences, India Section B: Biological Sciences*, 86(2), 253-260.

Other relevant references include:

Stirling, C. H., Andersen, M. B., Warthmann, R., & Halliday, A. N. (2015). Isotope fractionation of ²³⁸U and ²³⁵U during biologically-mediated uranium reduction. *Geochimica et Cosmochimica Acta*, 163, 200-218.

Brown, S. T., Basu, A., Christensen, J. N., Reimus, P., Heikoop, J., Simmons, A., ... DePaolo, D. J. (2016). Isotopic Evidence for Reductive Immobilization of Uranium Across a Roll-Front Mineral Deposit. *Environmental Science & Technology*, 50(12), 6189-6198.

Placzek, C. J., Heikoop, J. M., House, B., Linhoff, B. S., & Pelizza, M. (2016). Uranium isotope composition of waters from South Texas uranium ore deposits. *Chemical Geology*, 437, 44-55.

Reviewer #3 (Remarks to the Author):

Review of “Biogenic non-crystalline U(IV) revealed as major component in uranium ore: Implications for U ore genesis and mining”

by Amrita Bhattacharyya, Kate M. Campbell, Shelly Kelly, Yvonne Roebbert , Stefan Weyer, Rizlan Bernier-Latmani and Thomas Borch

Manuscript Summary and Recommendation:

I reviewed the manuscript by Bhattacharyya and coauthors with great pleasure. I was broadly aware of this study from conversations with Jim Clay and was excited to accept the review. The study documents the uranium bearing phases and associated biological and isotopic characteristics, from a roll front uranium deposit in the Powder River Basin (Wyoming, USA). The main finding of the paper is that most of the uranium is not hosted in the predicted U minerals such as uraninite, coffinite and pitchblende but in short-ordered U-O forms referred to hereafter as non-crystalline U(IV). Moreover the manuscript asserts, based on the $^{238}\text{U}/^{235}\text{U}$ isotopic ratios that the primary mechanism of U reduction is biologically mediated.

The finding of non-crystalline U(IV) as the primary solid phase of U in sandstone hosted U deposits is both novel and will be of interest to many researchers interested in U fate and transport in both environmental science and as a geochemical proxy in the earth sciences. Moreover, the findings that most of the ore is formed by microbial U reduction is also an interesting and important finding. I don't necessarily agree with the conclusions of the paper but I think the interpretations are reasonable and are an important contribution to the field aforementioned fields.

I am not an expert in the spectroscopic techniques used by the researchers, however the supporting sequential U extractions bolster the authors' claims that most of the U does not occur as uraninite or similar minerals. My main issue with the manuscript, as presented, includes the extrapolation of a single sediment core to a generalized model for the formation of similar U ores. Relatedly, there are several gaps in the logical argument including the earlier identification of U minerals at the same field site¹ and reconciling the earlier reported ages of the deposit with the supposedly labile nature of non-crystalline U(IV)². Finally, there is insufficient description the basic properties of the core including no petrographic data, no U concentrations and the U isotope data do not appear in a data table. It is my opinion that this paper is suitable for *Nature Communications* but will be improved if the above points are addressed. Below I attempt to detail and justify my concerns to help the authors improve the manuscript.

Substantive Comments:

Data and data presentation: Whether in the main paper or in the supplement there really should be a description of the actual core samples and the core overall. The XRD data are a start but I cannot understand why there are no petrographic or SEM images of the core material selected for further study. Moreover, why are there percentage-based data for the U phases but the concentrations of U for the ore are also not published? It is quite common in roll fronts to have dramatic changes in the concentration of U over 0.25 m intervals. It is unclear from the presented data whether any of the discussed mineralogy or other features are at all related to the total U concentration. Since the U concentration is a by-product of the U isotopic method these data should be reported. Alternately, I assume that there is a down hole gamma log for the core which could also be used to describe the core and report an estimated ore grade. As a very specific request I think that the actual sand (usually noted with a letter designation such as O, K or M) should be included in the description as well.

I could not find the U isotope data in the manuscript or supplement. They should be included for the benefit of the community at a minimum. I am also a bit confused by the choice of how to present the isotopic data. Instead of color-coding the symbols why not plot the depth on the Y-axis? Also the crust is not a singular value, there is some uncertainty and it should be represented as such³.

I am curious why only the <2mm fraction of the sediment was used? Is there any estimate how much of the total U was in the >2mm fraction? In my experience there is significant U coated on quartz and some of the coarsest sediments have high U concentrations.

Data interpretation: It seems to me that one of the outstanding issues raised by the manuscript is how roll fronts are stabilized if the primary solid phase is not a thermodynamically stable mineral. For example, how do the authors reconcile the U-Pb age from the introduction¹ with their observations? Do they think that non-crystalline U(IV) persists in nature for millions of years? If not, then are the roll fronts actually stable or are they migrating at some measurable rate? The non-crystalline U(IV) presumably has very high surface area compared to uraninite/coffinite mineral grains and thus must be susceptible to greater dissolution rates?

I am confused by the fraction of biogenic uraninite reported. Again, the spectroscopy is not my area of expertise so am not criticizing this result I just don't understand it. Is this really uraninite or is it some short ordered U-O solid? My understanding was that the EXAFS is not uniquely capable of distinguishing biogenic and inorganic uraninite from one another. If, on the other hand, the authors are using the isotopic results to infer that the uraninite component in their spectroscopic data is biogenic I think this is flawed. Even if one accepts the Stylo et al results at face value, you cannot know the fractionation direction or magnitude *a priori* without the coexisting fluid. Moreover, if the precipitation of U at the redox front is nearly quantitative then no meaningful fractionation is recorded in the solid phase. That said if the authors (and other reviewers) think the spectroscopic data

alone are evidence of the biogenic nature of reduction then I think its fine to use the isotopic data to generally support this conclusion.

In the U isotope section there are some logic gaps that need to be addressed. For example, if you apply the logic of this paper and the referenced Stylo paper then the $\delta^{238}\text{U}$ should be fractionated from the "crust" value by ~ 1 per mil. Since the largest observed fractionation is ~ 0.5 per mil this would imply that $\sim 50\%$ or so of the U reduction must be inorganic in nature (assuming no fractionation for inorganic reduction), but this conclusion is not supported by the data. As I mentioned above as you approach quantitative reduction this does not hold true, but this is the danger in applying closed system experimental results to an open system.

Minor Issues

Line 150: To be very clear the Stylo et al results for inorganic U reduction are inconsistent with the NFSE theory, which predicts isotopic fractionation between inorganic U(VI)-U(IV). This in no means invalidates the results, in fact, I am a strong advocate for using the experimental and observational data, theory will evolve to interpret these data. The language in this section should just be refined to be consistent with the actual theory and experimental results.

Line 165: The presence of bacteria capable of reducing U is not evidence of U reduction by bacteria. Microbial community analysis measures the identity and amount of microbes, not activity⁴.

Finally, I want to reiterate that I enjoyed reviewing this paper and look forward to seeing it published in the near future.

Shaun T. Brown
Berkeley, CA

1. Santos, E. S. & Ludwig, K. R. Age of uranium mineralization at the Highland Mine, Powder River basin, Wyoming, as indicated by U-Pb isotope analyses. *Economic Geology* **78**, 498–501 (1983).
2. Cerrato, J. M. *et al.* Relative Reactivity of Biogenic and Chemogenic Uraninite and Biogenic Noncrystalline U(IV). *Environmental Science & Technology* **47**, 9756–9763 (2013).
3. Noordmann, J., Weyer, S., Georg, R. B., Jöns, S. & Sharma, M. $^{238}\text{U}/^{235}\text{U}$ isotope ratios of crustal material, rivers and products of hydrothermal alteration: new insights on the oceanic U isotope mass balance. *Isotopes in Environmental and Health Studies* 1–23 (2015). doi:10.1080/10256016.2015.1047449
4. Gallegos, T. J., Campbell, K. M. & Zielinski, R. A. Persistent U (IV) and U (VI) following in-situ recovery (ISR) mining of a sandstone uranium deposit, Wyoming, USA. *Applied ...* (2015).

RESPONSE TO REVIEWER COMMENTS

The reviewers' comments helped to improve the quality of our original manuscript, and we greatly appreciate their efforts and input. This response is organized to address each reviewer's comments in turn. Unless otherwise indicated, line and page numbers refer to the revised manuscript.

RESPONSE TO REVIEWER 1:

REVIEWER COMMENT: L72: *what is an "in-situ recovery u mine"? should be defined for non-specialist.*

RESPONSE: In situ leaching (ISL), also known as solution mining, or in situ recovery (ISR) in North America. Uranium (U) in situ recovery (ISR), is a solution-based mining technique which is used to extract U from low grade ore-deposits by pumping down oxidants like oxygen/H₂O₂ and CO₂ fortified native groundwater via injection wells. The oxygen in the water oxidizes the U^(IV) into its soluble U^(VI) form. Bicarbonate promotes the formation of highly stable and mobile U^(VI)(hydroxy)carbonate ions. The uranium bearing water is then pumped back to the surface, where the uranium is extracted through an ion exchange process. Consequently there is little surface disturbance and no tailings or waste rock generated during ISR U mining operations as compared to the traditional open-pit mining ones.

CHANGE: ISR mining was described briefly on lines 55-56 and again within the Methods section (Lines 239-246).

REVIEWER COMMENT: L77 *"call into question the prevailing model for uranium ore formation and the U speciation therein and warrant the re-evaluation of the paradigm in order to optimize ore exploration and exploitation, as well as mine restoration." This is grand claim, and one wonders whether this is supported by analysis of the restricted number of samples (which is not unequivocal; see below)*

RESPONSE: We agree with Reviewer 1 that this statement is bold. However, we have full confidence in the experimental evidence we have presented in the ms. The concern which Reviewer 1 has raised regarding the restricted number of samples is perhaps expected. However, we would like to mention that no study has been conducted at ISR facilities till date, that involve characterization and speciation of undisturbed roll-front deposits by X-ray absorption spectroscopy, 16S rRNA, and isotope analysis, due to the difficulty in getting access to these sites, and anoxic sample preservation and analysis. Also, the data we have provided are statistically significant based on the statistical analysis that we have conducted for the samples studied.

CHANGE: We have toned down the statement and made some modifications within the text in the introductory section and under the "Implications for Uranium Ore Genesis and U mining" section. The modified text has been included in lines 79-85 and Lines 205-232.

REVIEWER COMMENT: *L109 “Tab. 1a” in the main body of the text, write out in full. Also fitting needs the “end-member” spectra identified in the caption.*

RESPONSE: We agree.

CHANGE: The above comments have been addressed. The word “Table” has been spelled out in Line 117. The caption in Lines 648-650 has been expanded to include the “end-members” so that it now reads: **Table 1a.** U speciation from EXAFS analysis and percentage of U^(IV) and U^(VI) within given sample depths (m-bgs) based on linear combination fitting of U EXAFS spectra with three phases; biogenic UO₂, non-crystalline U(IV), and U(VI) associated with Fe(oxy)hydroxide-like phases.

REVIEWER COMMENT: *L111 “Biogenic uraninite” how is the uraninite identified as biogenic in origin?*

RESPONSE: Biogenic uraninite is defined as the end product of uranium reduction mediated by environmentally relevant Gram-positive and/or Gram-negative bacteria. These consist of short-range ordered, amorphous nano-particulate uraninite (Schofield et al., 2010). Structurally, the U L_{III}-edge EXAFS spectra for biogenic UO₂ are found to be a best fit with half the number of second-shell uranium neighbors compared to crystalline inorganic uraninite, and no oxygen neighbors were detected beyond the first shell around U^(IV) in the biogenic UO₂.

CHANGE: The information has been introduced in line 66 and developed further in Lines 305-308.

REVIEWER COMMENT: *L114 “There was no evidence of coffinite or other U(IV) ore minerals present.” Therefore, the samples analyzed would seem to be significantly different to samples characterized previously as “Based on X-ray diffraction (XRD), a previous study identified pitchblende, uraninite and coffinite as the main U-bearing minerals in the ore³⁰.” One wonders how representative the samples in this study were of those studied previously, and how heterogeneous the mineralogy of the site is (which clearly impacts on extrapolation; see L77 comment).*

RESPONSE: Reference 30 cited in this line discusses the mineralogy of sediment samples collected from the Highland region of Southern Powder River basin in Wyoming, about 10 km away from where Smith Ranch Highland, our study site is located. Hence it is justified to say that the bulk mineralogy of the sediments will be closely representative of the samples used in our study, though the heterogeneous nature of these sediments cannot be ignored. The study discussed in Ref 30 used powder X-ray diffraction technique of selected U-containing grains determined by reflectance microscopy to identify the highly crystalline U-containing minerals present at the study site. This technique is not capable of identifying short range order structures of non-crystalline U(IV) as mentioned in Ln 86 even if they were present at the study site. On the

other hand, we did XRD on our bulk sediments (Table S2) which is used to characterize U minerals on the bulk <2 mm fraction, which would only detect U-containing phase at >1-2% by weight. Because the ore zone had <1% U, we would not expect to detect these phases by XRD, but rather used XRD to characterize the overall mineralogical composition of the samples. XAS is a superior technique for targeting U oxidation state and binding environment in a bulk samples when U concentrations are <1%. Moreover, we used bulk U EXAFS to identify the short range ordered U(IV) species whose presence have been previously overlooked in roll-front deposits. In summary, the tools used to identify coffinite and other U(IV) ore minerals in the previous investigation of this ore are not representative of the bulk composition of the ore, while the tools we use, namely bulk XAS and bulk XRD, are.

CHANGE: This point is now mentioned in line 94 where ‘individual grains’ are indicated are the source of the results of the previous study to clarify the differences between our method and the referenced study.

REVIEWER COMMENT: L118 “While the model invokes contributions from U-U pair correlations indicative of U(IV) minerals such as uraninite” is discounted, but could have been strengthened by other mineralogical analyses e.g. HRTEM (which would also help strengthen the overall mineralogical analysis of the sediments). In my opinion, too much hangs on the EXAFS analyses, which is a little difficult to follow and seems to have been presented rather selectively.

RESPONSE: It is true that high-resolution transmission electron microscopy (HRTEM) is an excellent imaging tool, and that it can be applied to natural and heterogeneous samples like soil or sediments. A member of our team (Bernier-Latmani) has extensive experience using HRTEM and STEM to identify specific mineral and amorphous species in sediments. However, carrying out TEM analyses would not resolve the question that the reviewer brings up. Even if we find evidence of UO₂ in the sediment, it does not mean that this mineral represents a significant fraction of the U in the sample. The variation between SEM, fission track, and bulk U results was demonstrated in post-mining samples from Smith-Ranch Highland in Gallegos et al., 2015. This is an inherent limitation of microscale tools. They provide little insight into the abundance of the species identified. This is where bulk EXAFS is most powerful: it provides a unique overview of the speciation of a given element at the bulk scale. This is exactly what is needed for this study, which is why it was selected as the main tool.

The fact that EXAFS analyses are difficult to follow is a separate issue and is one we have tried to address by providing a short primer on U EXAFS in the supplementary information in Lines 726-735.

Examples of the successful use of U EXAFS to identify the bulk speciation of U in natural samples include Mikutta et al. (2016), Bargar et al., 2013, Gallegos et al. (2015) and Wang et al. (2013).

CHANGE: We have addressed this comment by providing a short primer on U EXAFS in the supplementary information in Lines 726-735.

REVIEWER COMMENT: *L157, the authors imply a high abundance of uranium-reducing organisms in the samples (“0.1-31%”) but this is not really true of those that are known to reduce U(VI). The authors note these include “Pseudomonas, Clostridium, and Geobacter (Supplementary Tab. S4); other core samples from the same mine site also contained Desulfovibrio and Shewanella.”*

RESPONSE: We were not intending to imply high abundance of organisms, simply that they were present to some degree in the samples. As reviewer 3 also pointed out, presence does not necessarily imply activity, so we were asked to make this point more clear in this section.

CHANGE: We have added text to this section as noted to address this issue in Lines 174-177.

REVIEWER COMMENT: *Pseudomonads are not normally associated with U(VI) reduction (and the reference cited to support this is on U(VI) reduction by Clostridium species). As the 31% abundance of the “uranium-reducing organisms” corresponds to the Pseudomonads in the 194 m bgs sample, it is misleading to note that “0.1-31%” of the sequence correspond to uranium reducing organisms.*

RESPONSE: There are known U(VI)-reducing Pseudomonads, such as *Pseudomonas putida* (Barton et al., 1996, Bacterial reduction of soluble uranium: the first step of in situ immobilization of uranium, Radioact. Waste Manag. Environ. Resour. 20: 141-151.) Several references that were in the reference section (including an excellent review by Wall and Krumholz 2006) were inadvertently removed from this citation in the text; our sincere apologies for this oversight.

CHANGE: The references have been corrected, and the Barton et al 1996 paper has been added to the references. These references include examples of known U(VI) reducers pertaining to the genera *Pseudomonas*, *Clostridium*, *Geobacter*, *Desulfovibrio*, and *Shewanella*.

REVIEWER COMMENT: *Shewanella species, which feature in the discussion (due to the work on non-crystalline U(IV) formation by this model organism) are not shown in table S4 so are presumably at extremely low abundance (and probably not relevant).*

RESPONSE: *Shewanella* species were found in another core taken from the same mine site post-restoration, and analyzed using the identical DNA sequencing techniques (including primers) as the results presented. These results were previously published in Gallegos et al 2015. We can appreciate that this may be confusing in this context.

CHANGE: We have eliminated the direct reference to the community as published in Gallegos et al 2015.

REVIEWER COMMENT: *Interestingly, Geothrix is one metal-reducing organism which was detected at relatively high levels in the samples (at 18.6% in the 188 m bgs sample) but it is not discussed at all. One could argue that it could potentially play a role in uranium reduction, although little work has been done on this organism in this context but it has been associated with sediments undergoing U(VI) bioreduction previously in other studies.*

RESPONSE: We agree that Geothrix is likely a U(VI)-reducing organism, but to our knowledge, there is no paper published that conclusively demonstrates enzymatic U(VI) reduction by Geothrix (as opposed to an indirect mechanism such as Fe(III)-reduction). As a result, we chose not to specifically reference it as a potential U-reducer in this context without more conclusive demonstration of its U reduction, as that affects the isotopic interpretation.

CHANGE: None.

REVIEWER COMMENT: *In general however, it is difficult to describe the microbial communities detected as those expected in a zone of substantial metal/uranium bioreduction as implied in the manuscript (especially for sample 194 which seems to be the only sample that has been analysed for U speciation, isotopes and microbial composition).*

RESPONSE: We agree that it is important to not overstate the link between the current microbial community composition and the presence of biogenic U(IV) in the ore zone. We merely suggest that the presence of potential U(VI) reducing organisms is in accordance with the stronger isotopic, spectroscopic, and extraction data.

CHANGE: We have clarified this in the “Microbial communities present within ore deposit” section in Lines 172-204.

REVIEWER COMMENT: *L188, The authors note that “These results indicate that roll-front deposits are heterogeneous and contain microenvironments where either abiotic or biotic U reduction can occur. Nonetheless, this study clearly shows that biotic reduction is important and maybe the dominant U reduction mechanism in this ore body. The long-term stability of non-crystalline U(IV) is not known, but these data suggest that it may be more stable under reducing environmental conditions than previously suggested.” Although I agree about the heterogeneity, the comments about the dominant mechanism in this ore body do not seem adequately supported, and the comments about long-term stability seem overly speculative based on this study; just as likely is that the authors could have sampled a very dynamic redox system with recently reduced/precipitated U(IV)?*

RESPONSE: We concur with Reviewer 1’s comment that it is possible that we have sampled a dynamic redox system. But at the same time, since the majority of the sediments analyzed hint at the formation of biogenic U(IV) species it is safe to say that it is not overly speculative to

suggest that biotic reduction is the dominant mechanism. As our U EXAFS data suggest, it is possible that the reduced U(IV) species within these deep sediments get stabilized by binding to C-containing oxalate-like functional groups. There is a dearth of information regarding the long-term stability of non-crystalline U(IV) under reducing conditions. The best information comes from wetlands (Wang et al., Nat. Comm., 2013 and Mikutta et al., ES&T 2016) and suggests stability within the time frame of these systems (<10,000 years). On a longer time scale relevant for roll-front deposits, it is difficult to comment except to say that the wetland systems vastly exceed the expected stability of non-crystalline U(IV) based on laboratory experiments. Thus, it is conceivable that non-crystalline U(IV) is rather stable under low-flow, reducing conditions. Laboratory studies have also shown that surface associated organic matter tend to decrease the reactivity of biogenic nano-particulate U(IV) species (Fletcher et al,2010) .

CHANGE: We have modified the “Implication section” to include the two references cited above.

REVIEWER COMMENT: *L264 “Bulk U LIII-edge (17166 eV) extended x-ray absorption fine structure (EXAFS) data were collected for sediments with sufficient U concentrations.” What is limit of sensitivity for these analyses, and how does this relate to ore concentrations that are exploited in the body.*

RESPONSE: We agree that the detection limits were not adequately addressed.

CHANGE: We added the following section “EXAFS limits” to the Supplementary Information section in Lines 737-749.

EXAFS limits:

EXAFS is a unique tool for determining the U speciation within the sediments because the signal originates only from the U atoms within the samples with no competing signal from other crystalline or non-crystalline components. There is a minimum amount of U needed in the sample to obtain a measureable signal. This limit depends on the details of the measurement for which we estimate that limit to be approximately 100mg/kg. There is also a limit to the resolution of EXAFS to determine minor phases within a sample. In general this limit is about 5 to 10% depending of the specific nature of the phase. The EXAFS signal is a sum of the signals from all the U atoms within the sample. In general if 50% of the atoms have one type of neighbor, the a signal from 0.5 an atom on average can be detected. For example, if the sample contains 50% of UO₂, and each U atom in this phase is coordinated by 12 U atoms, then the EXAFS spectrum would contain a signal from 6 U atoms which is easily detected. Less than 5 mol% of all the U atoms in the sample are part of UO₂ would give an average signal of 0.6 U atoms which would be at the edge of our detection limit.

REVIEWER COMMENT: *Looked at holistically, the data are confusing and not always consistent with the hypothesis proposed. For example, the 196 m sample, is described as 88%*

non-crystalline U(IV) from the EXAFS, but from the $\delta^{238}\text{U}$ values it is identified as “abiotic” in origin.

RESPONSE: We are not proposing that biotic reduction is the only reduction mechanism operating here, although the most dominant one as found from our study. We put forward evidence, which suggests that this is a redox dynamic and heterogeneous system where there is interplay of biotic and abiotic reactions occurring simultaneously. Previous work documenting the possibility of the formation of non-crystalline U(IV) by abiotic reduction has been documented in Lines 191-204. What we want to highlight in this study is a) the high percentage of non-crystalline U(IV) found within these roll-front deposits in contrast to the commonly reported crystalline uraninite and b) that biotic reduction can be a dominant mechanism resulting in the formation of non-crystalline U(IV) species within roll-front deposits – historically roll-front formation studies have mostly been focused on abiotic U reduction mechanisms.

CHANGE: The information has been incorporated in Lines 155-171 and Lines 191-204.

REVIEWER COMMENT: *Also, it is concerning that the clear peaks expected for U-U in the EXAFS spectra are discounted; they seem quite pronounced in Figure 3 (at about 3.5 Å). This feature seems to be present even in the sample from 194.4 m-bgs, but in the legend in S4 it is noted as absent (although the plots are not shown in S4).*

RESPONSE: There is a signal in the EXAFS spectra in Figure 2 at about 3.5Å, but this signal is not due to U-U, but is rather due to C and O neighbors. The U-U signal due to uraninite is much larger than the signal in figure 2 as shown in now figure S3 (formerly Fig S4).

CHANGE: We further explained in the text related to Figure S3 to be specific about the large amplitude of the U-U signal.

REVIEWER COMMENT: *For S-4 it would be useful to see the plots from all the samples. I would question that at L120 “While the model... invalidating...”, if you look at the C2 fits, the errors are large (typically >50 %) and thus there is perhaps a more reasonable view that there is significant uncertainty in this area within the EXAFS due to the complexity of the fits and really whether it is C2 or U-U is unclear although the C2 looks perhaps slightly better fitting. I am not sure you can be as strong as saying U-U is invalidated. I wonder what happens if you exclude the C2 and introduce the U – U as the main fit – does this give realistic values?*

RESPONSE: We agree with the reviewer that the uncertainties are large and we do not hide this. But the fact that the uncertainty in the U-U coordination number (CN) is larger than the CN number indicates that the model can describe the data just as well without the U-U signal. The uncertainty in C2 is smaller than the CN number and thus indicates that this signal is required to have a model of this quality. So the data are clear. U-U is not required, but could persist at a low level, while C2 is required.

CHANGE: We added two more sentences to the discussion in Lines 132-139 the text to address this difference. Figure S3 (formerly S4) shows the data for all depths with the *k*-weight analysis.

REVIEWER COMMENT: *“While the model invokes contributions from U-U pair correlations indicative of U^(IV) minerals such as uraninite, the uncertainty in the U-U coordination number (CN) is as large or larger than the CN at all depths, invalidating significant contributions from uraninite (Tab. 1b and Fig. 2C).*”

RESPONSE: The U-U signal is not required to have the same quality of fit between model and data. The uncertainty in C2 while large is smaller than the CN, indicating that the model requires this signal to maintain the quality of fit.”

CHANGE: The text has been edited to clarify the statement in Lines 126-132.

RESPONSE TO REVIEWER 2:

REVIEWER COMMENT: *Whilst I feel that this manuscript is well written and topical, and the data appear robust, the role of bacterially-mediated reduction of U(VI) to U(IV) in low temperature U ore deposits has been identified in numerous other studies (many of which have not been cited in this manuscript), and U biomineralisation is well established as an important mechanism through which the low temperature uranium ore deposits form. An earlier study by Fletcher et al. (2010) (cited) presented EXAFS data from U reduction experiments using live *Desulfotobacterium* cultures, and showed that the U(IV) product was a phase or mineral composed of mononuclear U(IV) atoms closely surrounded by light element shells, which is likely the result of inner-sphere bonding between U(IV) and C/N/O- or P/S-containing ligands, such as carbonate or phosphate. This previously published finding is similar to that of this study. Thus, my primary concern with this manuscript is the lack of a particularly original and novel story.*

RESPONSE: We would like to differ with this comment regarding the novelty of the study. As clearly described in our manuscript, our study is the first to report non-crystalline U(IV) under native undisturbed conditions within a roll-front deposit. The aforementioned studies target microbial U(VI) reduction within a laboratory setting or bioremediation of U(VI) within contaminated shallow aquifer systems, which are very different from ore formation in roll front deposits. Our study attempts to characterize U naturally present in undisturbed roll front deposits and therein lies our novelty as rightly pointed out by Reviewer 3.

CHANGE: We have included text within the ms to highlight the novelty in Lines 79-85. We have also clarified that these results are specific to roll-front deposits (as opposed to all low temperature U ore deposits); the novelty of this work is still significant because roll-fronts are a very important type of deposit economically and environmentally.

REVIEWER COMMENT: *A second concern is that many of the geochemical techniques were not applied to the same samples e.g. the bacterial assays were not conducted on the same samples as the isotopic measurements/EXAFS samples - so it is not possible to unequivocally say that the isotopic signatures are a result of biological/abiotic reduction without knowing the exact bacteriological species present. This study would benefit in general from additional isotopic measurements, particularly on samples that where it is suggested that abiotic mineralisation is the dominant U reduction mechanism. Whilst this might not be possible, measurement of the isotopic signatures for each of the sequential extractions would be interesting, particularly to see if the fractions associated with Fe/Mn oxides give a ~0 to 0.2 ‰ isotopic shift as observed by Brennecke, G. A., Wasylenki, L. E., Bargar, J. R., Weyer, S., & Anbar, A. D. (2011). Uranium Isotope Fractionation during Adsorption to Mn-Oxyhydroxides. Environmental Science & Technology, 45(4), 1370-1375.*

RESPONSE: We attempted to extract DNA from all of the core samples, but this is a very low biomass system and amplification was not possible on many of the samples. Similarly, it was not possible to collect bulk U –EXAFS data on some sediments since the U concentrations were not high enough (<50 mg kg⁻¹) on some sediments to collect high quality U EXAFS data. Unfortunately, we do not have the residues of the sequential extractions left to measure the isotopic signatures. Measuring the isotopic signature of U extracted from Fe/Mn oxides would be helpful to confirm the validity of the extraction but would not provide any information on the biological origin of the reductive process.

CHANGE: None.

REVIEWER COMMENT: *It also would be useful to know where in the deposit the samples come from - e.g. high grade/low grade/unmineralised, up-gradient/down-gradient of the roll-front, within the limb/nose etc and any U concentrations of the samples.*

RESPONSE: The core was collected from a roll-front present in the ore zone. The total U concentrations for the sediments as obtained from HF digests are reported in the figure caption of Fig 1. We have attached logs and ore grade data for the hole from where the core was collected within the “Extended Data” section. We have also attached a map showing the location of the core in the Mine Unit 3 extension (Fig S2). The hole from which our core was taken was cased and completed, and the log was done after casing. Only the gamma log is useful (red trace) because gamma radiation easily penetrates the casing. The log is attached at two different scales (1:50 scale and 1:610-670 scale) to get a closer view of the region with high activity. The “oregrade” pdf shows the estimated weight % uranium at various depths. Also, it is perhaps important to note that in one zone, the ore grade reaches 0.2%. Fig S2 shows the production mine unit with Monitoring Well 3-1 (MOW 3-1), where the core was taken, and the production and injection wells for that area of the ore body. The locations of these wells show us where we believe the roll front is located. Please note that none of these wells had yet started operating when the core was taken and represented complete native conditions. The gamma log indicates

that the ore zone from where the core was taken is, by mining standards, good ore quality and hence was a suitable choice for sampling of core.

CHANGE: Additional information has been incorporated in Lines 239-246.

RESPONSE TO REVIEWER 3:

REVIEWER COMMENT: *Data and data presentation: Whether in the main paper or in the supplement there really should be a description of the actual core samples and the core overall. The XRD data are a start but I cannot understand why there are no petrographic or SEM images of the core material selected for further study.*

RESPONSE: We have NOW attached logs and ore grade data for the hole from where the core was collected within the "Extended Data" section. We have also attached a map showing the location of the core in Mine Unit 3 extension within the Supporting Information section. The hole from which our core was taken was cased and completed, and the log was done after casing. Only the gamma log is useful (red trace) because gamma radiation easily penetrates the casing. Please ignore the resistivity and spontaneous potential tracings on the graphs. The log is attached at two different scales (1:50 scale and 1:610-670 scale) so we can get a closer view of the region with high activity. The "oregrade" printout shows the estimated weight % uranium at various depths. Also, it is perhaps important to note that one ore grade reaches 0.2%. The map shows the production mine unit with Monitoring Well 3-1 (MOW 3-1), where the core was taken, and the production and injection wells for that area of the ore body. The locations of these wells show us where we believe the roll front is located. Please note that none of these wells had yet started operating when the core was taken and represented complete native conditions. The gamma log indicates that the ore zone from where the core was taken is, by mining standards, good ore quality and hence was a suitable choice for sampling of core.

A post-mining analysis on downgradient sediments by Gallegos et al 2015, shows substantial heterogeneity in similar sediment type, and includes SEM and fission track analyses. SEM and fission track are generally not quantitative and are not always representative of the bulk composition, which was the goal of our analyses. We needed quantitative bulk analyses to determine dominant processes at work in the ore zone. Hence, bulk U-EXAFS data was collected to comprehensively probe the oxidation state and chemical coordination environment of amorphous and non-crystalline U. Collectively, wet chemical, spectroscopy, 16S rRNA microbial and U-isotope data provide complementary sets of evidence to test our hypothesis.

CHANGE: Additional information has been incorporated in Lines 251-271.

REVIEWER COMMENT: *Moreover, why are there percentage-based data for the U phases but the concentrations of U for the ore are also not published? It is quite common in roll fronts to have dramatic changes in the concentration of U over 0.25 m intervals. It is unclear from the*

presented data whether any of the discussed mineralogy or other features are at all related to the total U concentration. Since the U concentration is a by-product of the U isotopic method these data should be reported.

RESPONSE: The y-axis on Figure 1 represents the concentrations of U in mg kg⁻¹. The total U concentrations of each sediment depth as obtained from the HF digestions are also provided within the Fig caption. ²³⁸U/²³⁵U ratios and corresponding δ ²³⁸U values (in per mil) for sediment samples are presented in Table S6. Thus, we believe that we have included all of the information requested by the reviewer.

CHANGE: Table S4 has been incorporated.

REVIEWER COMMENT: *Alternately, I assume that there is a down hole gamma log for the core which could also be used to describe the core and report an estimated ore grade.*

RESPONSE: We fully agree with the reviewer.

CHANGE: We have attached the down-hole gamma log for the entire core in the Extended Data section.

REVIEWER COMMENT: *As a very specific request I think that the actual sand (usually noted with a letter designation such as O, K or M) should be included in the description as well.*

RESPONSE: K and M are arbitrary designations of sands that are only used internally within Cameco. By that, we mean they would not be recognized in the scientific literature or even by geologists from a different mining company and so were not included in the manuscript. At Smith Ranch, the geologists designated sands by depth using letters. On the Highland side of the property about 10 km away, exploration was done by other geologists who designated the same sands by numbers (“10” sand, “20” sand, etc.). Mine Unit 3 (where our core came from) is in the “O” Sand. We also have studies based on another location which is a post-mining site (Mine Unit 4) and were taken from the “M” sand, so they are from a different horizon below our core location.

CHANGE: None.

REVIEWER COMMENT: *I could not find the U isotope data in the manuscript or supplement. They should be included for the benefit of the community at a minimum. I am also a bit confused by the choice of how to present the isotopic data. Instead of color-coding the symbols why not plot the depth on the Y-axis? Also the crust is not a singular value, there is some uncertainty and it should be represented as such. I am curious why only the 2mm fraction? In my experience there is significant U coated on quartz and some of the coarsest sediments have high U concentrations.*

RESPONSE: We agree with Reviewer 3. Figure 3 has been revised and the $\delta^{238}\text{U}$ values have been plotted as a function of depth. The <2 mm fraction was chosen in order to allow for comparison of the same sample fraction among the different types of analyses, for instance, both U sequential extraction and bulk U-EXAFS require the sediments to be less coarse in order to ensure complete digestion during the extractions and to get a low signal/noise ratio during EXAFS data collection. HF digestions were also done on the <2mm fraction, but HF is a much stronger extractant which ensures a complete digestion of the samples. More importantly, **the <2 mm fraction indeed represents the majority of the sediment** (very little to no sediment was retained after sieving) within the sampled core including the very coarse (quartz) sand fraction (i.e., the 1 mm to 2 mm fraction). [ISO 14688 grades sands as fine, medium and coarse with ranges 0.063 mm to 0.2 mm to 0.63 mm to 2.0 mm. In the United States, sand is commonly divided into five sub-categories based on size: very fine sand (1/16 – 1/8 mm diameter), fine sand (1/8 mm – 1/4 mm), medium sand (1/4 mm – 1/2 mm), coarse sand (1/2 mm – 1 mm), and very coarse sand (1 mm – 2 mm).]

CHANGE: U isotope data has been included in the Supplementary Information section as Table S4. Figure 3 has been modified and now represents the $\delta^{238}\text{U}$ values plotted against depth. The crustal value is now shown as a box representing a "crustal range" which, according to Tissot and Dauphas (2015) and Noordmann et al. (2016), encompasses the $\delta^{238}\text{U}$ for most typical crustal rocks. On page 9 lines 274-276 we added the following text after "2-mm sieve" "(allowing the majority of the sediment to be characterized including "very coarse sand")."

REVIEWER COMMENT: *How do the authors reconcile the U-Pb age from the introduction with their observations? Do they think that noncrystalline U(IV) persists in nature for millions of years? If not, then are the roll fronts actually stable or are they migrating at some measurable rate? The noncrystalline U(IV) presumably has very high surface area compared to uraninite/coffinite mineral grains and thus must be susceptible to greater dissolution rates?*

RESPONSE: From our personal communications with Dr. Jim Clay, the principal scientist at Cameco we have learnt that the apparent ages of the roll fronts are somewhere in the vicinity of 2 to 2.5 million years. The age was determined using a sample from a roll front in the Highland mine which was a very high grade (19% U), calcite-cemented ore (Ludwig and Grauch, 1980). We suspect the roll front moves sporadically in bursts that start when a geological event such as an uplift occurs. In other words, we think the rate of movement is highly variable over time. It is inconclusive to determine whether the roll front in recent geological time is migrating at a measurable rate given the U-Pb data.

There is a dearth of information regarding the long-term stability of non-crystalline U(IV) under reducing conditions. The best information comes from wetlands (Wang et al., Nat. Comm., 2013 and Mikutta et al., ES&T 2016) and suggests stability within the time frame of these systems (<10,000 years). On a longer time scale relevant for roll-front deposits, it is difficult to comment except to say that the wetland systems vastly exceed the expected stability of non-crystalline U(IV) based on laboratory experiments. Thus, it is conceivable that non-crystalline

U(IV) is rather stable under low-flow, reducing conditions. Laboratory studies have also shown that surface associated organic matter tend to decrease the reactivity of biogenic nano-particulate U(IV) species (Fletcher et al., 2010).

CHANGE: The Implication section has been modified in the ms in Lines 205-232.

REVIEWER COMMENT: *I am confused by the fraction of biogenic uraninite reported. Again, the spectroscopy is not my area of expertise so am not criticizing this result I just don't understand it. Is this really uraninite or is it some short ordered U-O solid? My understanding was that the EXAFS is not uniquely capable of distinguishing biogenic and inorganic uraninite from one another.*

RESPONSE: Reviewer 3 is correct that biogenic uraninite is a short-range ordered U-O solid. The biogenic uraninite spectra used for the linear combination fitting of U-EXAFS data was from a sediment sample which was reduced via microbial reduction of uranyl under controlled laboratory settings in order to ensure complete bioreduction. The spectrum for this is used as a reference for biogenic uraninite in EXAFS analysis. It consists of short-range ordered, amorphous nano-particulate uraninite (Schofield et al., 2010). Structurally, the U L_{III}-edge EXAFS spectra for biogenic UO₂ are found to be a best fit with half the number of second-shell uranium neighbors compared to crystalline inorganic uraninite, and no oxygen neighbors were detected beyond the first shell around U(IV) in the biogenic UO₂ which is observed in the inorganic uraninite. We agree with Reviewer 3 that it would be difficult to distinguish biogenic from chemogenic uraninite in a natural sample, particularly in the presence of a high percentage of non-crystalline U(IV), as in our study, even though biogenic uraninite was used for the LCF analyses. We can infer the presence of non-crystalline U(IV) and possibly some minor contribution from uraninite which resembles the EXAFS spectrum from a short-range ordered U-O solid. Additionally, in order to test our hypothesis, the EXAFS spectrum of crystalline inorganic uraninite was compared with that of our sediment samples (Fig S5), which clearly showed the absence of this U(IV) species in our samples.

CHANGE: The definition of biogenic uraninite has been incorporated into the ms within Lines 305-308.

REVIEWER COMMENT: *If, on the other hand, the authors are using the isotopic results to infer that the uraninite component in their spectroscopic data is biogenic I think this is flawed. Even if one accepts the Stylo et al results at face value, you cannot know the fractionation direction or magnitude a priori without the coexisting fluid. Moreover, if the precipitation of U at the redox front is nearly quantitative then no meaningful fractionation is recorded in the solid phase. That said if the authors (and other reviewers) think the spectroscopic data alone are evidence of the biogenic nature of reduction then I think its fine to use the isotopic data to generally support this conclusion.*

RESPONSE: We agree with Reviewer 3 that knowledge of the fluid composition is important to precisely predict isotope fractionation during U reduction. The important point, however, is that it is difficult to generate heavy U isotope signatures without the mechanism of biogenic U reduction involved. Assuming, e.g. an open system from which isotopically heavy U is extracted from the fluid by biotic U reduction, the remaining fluid would become light (it can become very light, e.g. Murphy et al., 2014; Basu et al., 2015). This may result in low $\delta^{238}\text{U}$ values, even if U isotope fractionation during bio-reduction towards higher $\delta^{238}\text{U}$ occurred. Accepting that abiotic reduction results only in very minor isotope effects, we may use the measured isotope fractionation between the $\delta^{238}\text{U}$ of the samples and that of the crust to infer a minimum contribution of biological reduction, as all known interfering processes would result in lower $\delta^{238}\text{U}$ of our samples. These are:

1) The fluids from which the U of our samples is extracted was already light as a result from previous biotic U reduction. This process may even result in lower $\delta^{238}\text{U}$ of the samples compared to the crust, although reduction was biological; 2) Some U isotope fractionation, towards lighter compositions, may occur in theory during U oxidative U mobilization which would also result in isotopically light fluids. However, according to Wang et al. (2015), "little U isotope fractionation occurs during U oxidation"; and 3) reduction may have been almost quantitative, resulting in little isotope fractionation (though still towards heavier compositions). Thus, even if we cannot evaluate every single sample with the isotope signature, we can tell that biotic U reduction is the dominant process for the investigated samples.

However, we would like to emphasize that multiple lines of evidence were used which point towards biogenic U including isotopes, XAS, and 16S rRNA analyses.

CHANGE: Points 1 and 2 have been included in the text in Lines 155-160.

REVIEWER COMMENT: *In the U isotope section there are some logic gaps that need to be addressed. For example, if you apply the logic of this paper and the referenced Stylo paper then the $\delta^{238}\text{U}$ should be fractionated from the "crust" value by ~1 per mil. Since the largest observed fractionation is ~0.5 per mil this would imply that ~50% or so of the U reduction must be inorganic in nature (assuming no fractionation for inorganic reduction), but this conclusion is not supported by the data. As I mentioned above as you approach quantitative reduction this does not hold true, but this is the danger in applying closed system experimental results to an open system.*

RESPONSE: As we have mentioned before, in order to approach quantitative reduction, we need to know the fluid composition (please see the above response): 1) The crust gives just a value for the initial average fluid at best, assuming that no isotope fractionation occurred during U oxidation which may be assumed (Wang et al., 2015); 2) fluids become isotopically light during progressive U reduction in an open system. E.g. a value of +0.5 can be explained with an isotope fractionation factor epsilon of 1‰, either if the fluid was already light, i.e. had $\delta^{238}\text{U} = -0.5\text{‰}$ (relative to the crust) and assuming that only a very small amount was reduced, or, if the fluid was not light ($\delta^{238}\text{U} = 0.0$ relative to the crust) AND about 50% of the U was reduced in one

batch. Of course anything in between is possible as well. Thus, we can only say that, according to experimental findings (Basu et al., 2014; Stylo et al., 2015; Stirling et al., 2015), heavy U isotopes tell us that biotic U reduction “must” have been involved. It is typical to accept a range from -0.2 to -0.4‰, saying that about 90% of the rocks, representing the crust fall into this range [Tissot and Dauphas (2015) and Noordmann et al. (2016)].

CHANGE: Points 1 and 2 have been included in the text in Lines 155-160.

Minor Issues

REVIEWER COMMENT: *Line 150: To be very clear the Stylo et al results for inorganic U reduction are inconsistent with the NFSE theory, which predicts isotopic fractionation between inorganic U(VI)-U(IV). This in no means invalidates the results, in fact, I am a strong advocate for using the experimental and observational data, theory will evolve to interpret these data. The language in this section should just be refined to be consistent with the actual theory and experimental results.*

RESPONSE: We agree that abiotic reduction of U(VI) results in isotopic fractionation that is inconsistent with NFSE theory. However, given that the experimental data show this evidence, we used it as a tool to identify signatures of biotic U(VI) reduction in the ore. We have added a sentence in the manuscript to clarify the behavior during abiotic reduction is not consistent with theory. We would also like to cite here the paper from Wang et al. (2015) [Xiangli Wang, Thomas M. Johnson, Craig C. Lundstrom. Low temperature equilibrium isotope fractionation and isotope exchange kinetics between U(IV) and U(VI). *Geochimica et Cosmochimica Acta* 158 (2015) 262–275] which shows that if U(IV) and U(VI) coexist in a very acidic hydrochloric solution, U isotope fractionation is observed, consistent to that predicted by theoretical models. Potentially, in nature and experiments, U(IV) and U(VI) do not coexist in such a way that isotopic exchange can occur.

CHANGE: We have added a sentence in the manuscript to clarify the behavior during abiotic reduction is not consistent with theory in Lines 155-160.

REVIEWER COMMENT: *Line 165: The presence of bacteria capable of reducing U is not evidence of U reduction by bacteria. Microbial community analysis measures the identity and amount of microbes, not activity.*

RESPONSE: We completely agree with this statement, and have added text to this effect in the “microbial communities present within ore deposits” section (see also response to Reviewer 1).

CHANGE: The text has been modified in Lines 174-177.

REFERENCES:

1. Bargar, J.R. et al. Uranium redox transition pathways in acetate-amended sediments. *Proc. Natl Acad. Sci. USA* **110**, 4506–4511 (2013).
2. Barton, L.L., Choudhury, K., Thomsom, B.M., Steenhoudt, K. & Groffman, A.R. Bacterial reduction of soluble uranium: the first step of in situ immobilization of uranium, *Radioact. Waste Manag. Environ. Resor.* **20**, 141-151 (1996).
3. Basu, A. et al. Uranium isotopic fractionation factors during U(VI) reduction by bacterial isolates. *Geochim. Cosmochim. Acta* **136**, 100–113 (2014).
4. Fletcher, K.E. et al. U(VI) reduction to mononuclear U(IV) by *Desulfitobacterium* species. *Environ. Sci. Technol.* **44**, 4705–4709 (2010).
5. Gallegos, T.J. et al. Persistent U(IV) and U(VI) following in-situ recovery (ISR) mining of a sandstone uranium deposit, Wyoming, USA. *Appl. Geochem.* **63**, 222-234 (2015).
6. Ludwig, K.R. & Grauch, R.I. Coexisting coffinite and uraninite in some sandstone-host uranium ores of Wyoming. *Econ. Geol.* **75**, 296-302 (1980).
7. Mikutta, C., Langner, P., Bargar, J.R. & Kretzschmar, R. Tetra- and Hexavalent Uranium Forms Bidentate-Mononuclear Complexes with Particulate Organic Matter in a Naturally Uranium-Enriched Peatland. *Environ. Sci. Technol.* **50**, 19, 10465–10475 (2016).
8. Murphy, M.J., Stirling, C.H., Turner, S.P., Schaefer, B.F. & Kaltenbach, A. Redox-controlled fractionation of ²³⁸U/²³⁵U during low temperature uranium mineralisation. *Earth and Planetary Sci. Letters* **388**, 306-317 (2014).
9. Noordmann, J., Weyer, S., Georg, B. & Sharma, M. ²³⁸U/²³⁵U isotope ratios of crustal material, rivers and products of hydrothermal alteration: new insights on the oceanic U isotope mass balance. *Isotopes in Environmental & Health Studies* **52**, 141-163 (2016).
10. Schofield, E.J. et al. Structure of biogenic uraninite produced by *Shewanella oneidensis* strain MR-1. *Environ. Sci. Technol.* **42**, 7898–7904 (2008).
11. Stirling, C.H., Andersen, M.B., Potter, M. & Halliday, A.N. Low-temperature isotopic fractionation of uranium. *Earth Planet Sci. Lett.* **264**, 208–225 (2007).
12. Stylo, M. et al. Uranium isotopes fingerprint biotic reduction. *Proc. Natl Acad. Sci. USA* **112**, 5619-5624 (2015).
13. Tissot, F.L.H. & Dauphas, N. Uranium isotopic compositions of the crust and ocean: Age corrections, U budget and global extent of modern anoxia. *Geochim. Cosmochim. Acta* **167**, 113-143 (2015).
14. Wang, X., Johnson, T.M. & Lundstrom, C.C. Low temperature equilibrium isotope fractionation and isotope exchange kinetics between U(IV) and U(VI). *Geochim. Cosmochim. Acta* **158**, 262–275 (2015).

REVIEWERS' COMMENTS:

Reviewer #2 (Remarks to the Author):

I am happy that the authors have thoroughly addressed the point-by-point concerns of the reviewers, and I would be supportive of this manuscript being published with a few minor changes. My subsequent comments are only minor.

The figure detailing the $\delta^{238}\text{U}$ with depth showing the crustal range, it might be illustrative to also put a field on showing abiotic and biotic \pm low T redox ores on too.

A recent paper has come out supporting the argument here, presenting new $\delta^{238}\text{U}$ from adsorption experiments onto oxides (positive $\delta^{238}\text{U}$) versus biotic reduction (negative $\delta^{238}\text{U}$), and relating these to the U speciation using published XAFS data. It might be worth a cite?

(Dang, D.H., Novotnik, B., Wang, W., Georg, R.B., Evans, R.D., 2016. Uranium Isotope Fractionation during Adsorption, (Co)precipitation, and Biotic Reduction. *Environ. Sci. Technol.* 50, 12695–12704. doi:10.1021/acs.est.6b01459)

The addition of the gamma logs is useful, however compared to the rest of the high quality figures in the rest of the manuscript, they look quite low quality. Any chance of tidying them up, and indicating the depths and sample names at which the samples were taken?

Reviewer #3 (Remarks to the Author):

Review of “Biogenic non-crystalline U(IV) revealed as major component in uranium ore: Implications for U ore genesis and mining”

by Amrita Bhattacharyya, Kate M. Campbell, Shelly Kelly, Yvonne Roebbert , Stefan Weyer, Rizlan Bernier-Latmani and Thomas Borch

The manuscript submitted and revised by Bhattacharyya and coworkers on the nature and origin of U in mineralized sandstone “roll-front” deposits presents a new and unique take on the processes that form sandstone hosted U deposits. The manuscript is well written and the authors have addressed most of the reviewer comments in a satisfactory manner. The study certainly presents unique evidence in support of their interpretation and is sufficiently novel for publication in *Nature Communications*.

My main issue with the manuscript is transactional and not scientific. I still don't understand the unwillingness to provide the uranium concentration data for the sequential extractions in a useful data table. I think its unreasonable for readers of this article to have to digitize Fig 1 or comb through the text and try to calculate the U concentrations for various leach solutions. Moreover, putting the U concentrations for the total digests in the figure caption is not a particularly user-friendly way of reporting the data. I don't think the present format meets the sprit of the data availability policy for Nature publications.

A second issue is the fact that the highest concentration sample reported in the sequential extractions and isotope data has a much lower (~12% of peak U from gamma) U concentration than the reported ore grade from the gamma logs. This implies that the authors did not characterize a significant portion of the ore zone U. This “missing” U might be similar to the reservoirs reported in the manuscript but I don't see anyway to confirm this. At this point, I don't think this should stand in the way of publication but it is an interesting problem.

Minor point: the error bars on Fig 3 don't seem to match the data table. 2 data points have 0.1 and 0.09 per mil errors in the table but only one sample seems to have a similarly large error bar on the figure.

Shaun T. Brown
Berkeley

RESPONSE TO REVIEWER COMMENTS

The reviewers' comments have helped us to improve the quality of our original manuscript, and we greatly appreciate their efforts and input. This response is organized to address the few final comments from Reviewer 2 and 3. Unless otherwise indicated, line and page numbers refer to the revised manuscript.

Reviewer #2

REVIEWER COMMENT 1: The figure detailing the $\delta^{238}\text{U}$ with depth showing the crustal range, it might be illustrative to also put a field on showing abiotic and biotic±low T redox ores on too. A recent paper has come out supporting the argument here, presenting new $\delta^{238}\text{U}$ from adsorption experiments onto oxides (positive $\delta^{238}\text{U}$) versus biotic reduction (negative $\delta^{238}\text{U}$), and relating these to the U speciation using published XAFS data. It might be worth a cite? (Dang, D.H., Novotnik, B., Wang, W., Georg, R.B., Evans, R.D., 2016. Uranium Isotope Fractionation during Adsorption, (Co)precipitation, and Biotic Reduction. *Environ. Sci. Technol.* 50, 12695–12704. doi:10.1021/acs.est.6b01459)

Response to Reviewer Comment: *We agree with Reviewer 2 but at the same time we feel that it is a lot of data to include, which might make the figure, look cumbersome and distract from the main message. Instead, we have now included the reference cited by Reviewer 2 and another recent review article that has a comprehensive list of $\delta^{238}\text{U}$ values for lowT redox ores. These are Ref #s 43 and 44 in the revised ms [page 6 line 188].*

REVIEWER COMMENT 2: The addition of the gamma logs is useful, however compared to the rest of the high quality figures in the rest of the manuscript, they look quite low quality. Any chance of tidying them up, and indicating the depths and sample names at which the samples were taken?

Response to Reviewer Comment: *We agree. Thus, we have now provided an improved version of the gamma log in the revised Extended Data pdf.*

Reviewer #3

REVIEWER COMMENT 1: My main issue with the manuscript is transactional and not scientific. I still don't understand the unwillingness to provide the uranium concentration data for the sequential extractions in a useful data table. I think its unreasonable for readers of this article to have to digitize Fig 1 or comb through the text and try to calculate the U concentrations for various leach solutions. Moreover, putting the U concentrations for the total digests in the figure caption is not a particularly user-friendly way of reporting the data. I don't think the present format meets the spirit of the data availability policy for Nature publications.

Response to reviewer comment: *We have now added a new table (Supplementary Table 2 in the revised ms), which indicates the U concentrations (in mg kg⁻¹) based on ICPMS analyses of U sequential extractions and digests. However, we point out that the same data are now represented in BOTH Figure 1 and the NEW Supplementary Table 2*

REVIEWER COMMENT 2: A second issue is the fact that the highest concentration sample reported in the sequential extractions and isotope data has a much lower (~12% of peak U from gamma) U concentration than the reported ore grade from the gamma logs. This implies that the authors did not characterize a significant portion of the ore zone U. This “missing” U might be similar to the reservoirs reported in the manuscript but I don’t see anyway to confirm this. At this point, I don’t think this should stand in the way of publication but it is an interesting problem.

Response to reviewer comment: *We agree that it would have been nice with an even more comprehensive dataset than presented in this manuscript. However, due to the extensive amount of work and logistics related to analysis of this field site it was not possible for us to add more samples to our study. Since this is the first report of non-crystalline biogenic U(IV) within roll-front deposits we believe that the data presented will give rise to many additional studies to further explore the global significance of our findings.*

REVIEWER COMMENT 3: Minor point: the error bars on Fig 3 don’t seem to match the data table. 2 data points have 0.1 and 0.09 per mil errors in the table but only one sample seems to have a similarly large error bar on the figure.

Response to reviewer comment: *Thanks to reviewer 3 for pointing this out. We apologize that there was a mistake in the plotting. We have now fixed it in Fig 3.*